_Article_

EMBO
Molecular Medicine

# Intraductal xenografts show lobular carcinoma cells rely on their own extracellular matrix and LOXL1

George Sflomos[1] (iD), Laura Battista[1] (iD), Patrick Aouad[1] (iD), Fabio De Martino[1] (iD), Valentina Scabia[1] (iD), Athina Stravodimou[2] (iD), Ayyakkannu Ayyanan[1] (iD), Assia Ifticene-Treboux[2] (iD), RLS[3], Philipp Bucher[1], Maryse Fiche[3,4], Giovanna Ambrosini[1] (iD) & Cathrin Brisken[1,*] (iD)

## Abstract

Invasive lobular carcinoma (ILC) is the most frequent histological subtype of breast cancer, typically characterized by loss of E-cadherin. It has clinical features distinct from other estrogen receptor-positive (ER⁺) breast cancers but the molecular mechanisms underlying its characteristic biology are poorly understood because we lack experimental models to study them. Here, we recapitulate the human disease, including its metastatic pattern, by grafting ILC-derived breast cancer cell lines, SUM-44 PE and MDA-MB-134-VI cells, into the mouse milk ducts. Using patient-derived intraductal xenografts from lobular and non-lobular ER⁺ HER2⁻ tumors to compare global gene expression, we identify extracellular matrix modulation as a lobular carcinoma cell-intrinsic trait. Analysis of TCGA patient datasets shows matrisome signature is enriched in lobular carcinomas with overexpression of elastin, collagens, and the collagen modifying enzyme _LOXL1_. Treatment with the pan LOX inhibitor BAPN and silencing of _LOXL1_ expression decrease tumor growth, invasion, and metastasis by disrupting ECM structure resulting in decreased ER signaling. We conclude that LOXL1 inhibition is a promising therapeutic strategy for ILC.

**Keywords** extracellular matrix; lobular carcinoma; LOXL1; preclinical models; xenografts
**Subject Category** Cancer
See also: **KJ Kozma et al** (March 2021)

## Introduction

Breast cancer (BC) is a heterogeneous disease with 19 different histopathologic subtypes. Most carcinomas are classified as of no special type (NST). Among the special histological subtypes, invasive lobular carcinoma (ILC) is the most frequent, representing 10–15% of all cases (Tavassoli & Devilee, 2003; Lakhani, 2012). ILCs are characterized by small, non-cohesive, and non-polarized tumor cells with mild nuclear atypia that frequently grow as "Indian files" with little disruption of normal breast tissue architecture (Tavassoli & Devilee, 2003). ILCs are up to 95% estrogen receptor-positive (ER⁺), 60–70% progesterone receptor-positive (PR⁺), and tend to have a low proliferative index; all features usually associated with a good prognosis. However, ILCs are more often diagnosed at advanced stages than other ER⁺ BCs. The absence of a desmoplastic reaction and of microcalcifications in ILCs delay clinical detection by palpation and mammography. ILCs metastasize more frequently to the ovaries, the adrenal glands, and the meninges, and are less responsive to chemotherapy and tamoxifen than non-lobular ER⁺ BCs (de la Monte _et al_, 1984; Guiu _et al_, 2014; Christgen _et al_, 2016).

By global gene expression profiling (Perou _et al_, 2000) and PAM50 (Parker _et al_, 2009), which define four molecular subtypes, ILCs cluster with luminal A and B, and not HER2⁻ enriched, nor basal-like subtypes (Ciriello _et al_, 2015; Desmedt _et al_, 2016; Du _et al_, 2018). Cytogenetically, ILCs are more likely to be diploid than non-ILC tumors (Arpino _et al_, 2004). Loss of E-cadherin expression is a hallmark of ILC, seen in more than 90% of cases (Christgen _et al_, 2016).

Elegant genetically engineered mouse models carrying various mutations in conjunction with _CDH1_ deletion, mimic different aspects of the disease (Derksen _et al_, 2006; Boelens _et al_, 2016; Tenhagen _et al_, 2016; An _et al_, 2018) and of particular subtypes, such as the classic (Boelens _et al_, 2016) or inflammatory ILC (An _et al_, 2018). However, these models lack the high ER and PR expression characteristic of the human disease and are of limited value in mimicking the metastatic disease. Seven human ILC cell lines have been reported, all of which are difficult to grow _in vitro_ and more so as xenografts (Riggins _et al_, 2008; Sikora _et al_, 2014; Christgen & Derksen, 2015; Özdemir _et al_, 2018; Stires _et al_, 2018).

---

1   ISREC - Swiss Institute for Experimental Cancer Research, School of Life Sciences, Ecole Polytechnique Fédérale de Lausanne (EPFL), Lausanne, Switzerland
2   Lausanne University Hospital, Lausanne, Switzerland
3   Réseau Lausannois du Sein (RLS), Lausanne, Switzerland
4   International Cancer Prevention Institute, Epalinges, Switzerland
    *Corresponding author. Tel: +41 (0)21 693 07 81/sec: +41 (0)21 693 07 62; Fax: +41 (0)21 693 07 40; E-mail: cathrin.brisken@epfl.ch

We have recently shown that the notoriously low take rates of ER$^+$ BC xenografts of both cell lines and patient-derived tumor cells can be markedly improved by grafting the cells to the milk ducts of immunocompromised mice (Sflomos *et al*, 2016), a finding that extends to androgen receptor-positive, molecular apocrine BCs (Farmer *et al*, 2005; Richard *et al*, 2016). The ER$^+$ BC cell line, MCF7, faithfully recapitulates ER$^+$ BC development from *in situ* to metastatic disease when grafted this way (Sflomos *et al*, 2016) although the cell line is derived from a pleural effusion, i.e., late-stage disease, and has been passaged extensively *in vitro* (Soule *et al*, 1973). Here, we show that the ER$^+$ ILC-derived cell lines, MDA-MB-134-VI (MM134) and SUM-44 PE (SUM44), which are both functionally inactive for p53 (Reis-Filho *et al*, 2006; Wasielewski *et al*, 2006), recapitulate many features of the disease when xenografted intraductally and provide new robust models for this under-studied disease. Comparing lobular and non-lobular ER$^+$ HER2$^-$ PDXs, we identify a transcriptional program underlying the lobular-specific molecular characteristics and show that extracellular matrix (ECM) modulation is a tumor cell-intrinsic characteristic and targetable feature of ILCs.

## Results

### Intraductal xenografts of ILC-derived cell lines model human disease

To assess whether ILC-derived cell lines recapitulate the biology of human lobular carcinomas when xenografted to the mouse milk ducts, we tested the ER$^+$ lobular, E-cadherin- carcinoma-derived MM134, and SUM44 cell lines. To identify specific differences with non-lobular E-cadherin$^+$ ER$^+$ HER2$^-$ models, we compared them with the MCF7 and T47D cell lines. All cell lines were infected with RFP-luciferase 2 expressing lentivirus, selected by flow cytometry for RFP expression, and injected into the milk ducts of immunocompromised NOD.Cg-*Prkdc$^{scid}$* Il2rgtm1$^{Wjl}$/SzJ (NSG) females (Shultz *et al*, 2005) (Fig 1A). *In vivo* radiance measurements within a week of injection showed engraftment rates of > 83% for the lobular versus > 95% the non-lobular cell lines, respectively (Fig 1B). Measurements of *in vivo* radiance showed the ILC cells had lower growth rates than the non-ILC ER$^+$ HER2$^-$ cells (Fig 1C). The growth curves for MCF7 cells did not differ significantly from those for T47D cells, but differed significantly from the curves for MM134 ($P < 0.03$) and SUM44 ($P < 0.0005$) cells (Fig 1C). In particular, MCF7 and T47D cells showed faster growth over the first 4 weeks whereas the ILC cells grew at a more constant rate throughout the experiment. Five months after injection, glands xenografted with MCF7 or T47D cells were enlarged and hard on palpation, whereas MM134 and SUM44 xenografts were inconspicuous. Considering fold change in luminescence at endpoint, MM134 cells were only marginally different from MCF7 cells, whereas growth of the SUM44 xenografts differed by an order of magnitude (Fig 1D).

To analyze the growth of the different BC cell lines in the context of the mouse ductal system, we stained whole mounted mammary glands with carmine alum, a DNA dye, that reveals milk ducts with their densely packed epithelial cells while barely staining the fatty stroma. Stereomicroscopy of whole mounted uninjected glands from experimental animals showed a dichotomously branched ductal system with few side branches characteristic of young adult virgin mice (Fig 1E). Two months after injection, xenografted MCF7 cells caused widespread dilation of host ducts (Fig 1F) with tumor growth evenly distributed throughout the ducts but sparing the tips (Fig 1F). In contrast, MM134 (Fig 1G) and SUM44 (Fig 1H) cells homed to ductal tips and gave rise to grape-like structures reminiscent of the highly branched terminal ductal-lobular units (TDLUs) in the human breast (Fig 1G and H).

Thus, ILC cells have lower engraftment rates, grow slower than non-ILC, ER$^+$ HER2$^-$ BC cells, and colonize the tips of the murine ductal tree. Their ability to induce human-specific structures, TDLUs, points to distinct morphogenic properties of ILC cells.

### Metastatic spread of intraductal ILC xenografts

Within 2 months of intraductal engraftment, disseminated MCF7-luc2-GFP cells can be detected in distant organs by *ex vivo* bioluminescence (Sflomos *et al*, 2016), whereas there was no detectable

---

**Figure 1.   Intraductal xenografts of MM134 and SUM44 recapitulate lobular breast carcinogenesis.**

A      Schematic outline of the intraductal injection approach used to generate ILC xenograft models. ILC cells are transduced with RFP-luciferase expressing lentiviruses and injected into 3$^{rd}$ thoracic and the 4$^{th}$ inguinal milk ducts, highlighted in red, of 10-week-old NSG females. Tumor growth is monitored by bioluminescence imaging.

B      Bar graph showing percentage of engrafted mammary glands of non-ILC, ER$^+$ HER2$^-$; total number of glands injected with MCF7 ($n = 17$) and T47D ($n = 14$) and ILC, MM134 ($n = 38$) and SUM44 cells ($n = 47$). Black dotted line shows take rate 95%.

C      Graph showing *in vivo* growth of the different cell line xenografts. Curves represent mean log-transformed luminescence $\pm$ SEM of individual glands; MCF7 ($n = 17$), T47D ($n = 10$), MM134 ($n = 12$), and SUM44 ($n = 12$). Statistical testing for interactions group cell growth/ time relative to MCF-7 was carried out by applying mixed-effect linear models. *P*-values are associated with likelihood-ratio test statistics of the ANOVA function in R.

D      Box plot showing log$_{10}$ (radiance fold change) of individual glands 5 months after injection of MCF7, T47D, MM134, and SUM44 cells; $n = 17, 14, 38, 47$, respectively. Vertical lines outside the box end at maximum and minimum values, upper and lower borders of the box represent lower and upper quartiles, and line inside the box identifies the median. Statistical significance of difference to MCF7 xenografts determined by unpaired, Student's *t*-test, two-tailed, ***$P < 0.001$, n.s. not significant. *P*-values have been corrected for multiple testing by FDR.

E–H   Whole mount stereo-micrographs of carmine alum stained mammary glands either uninjected (E), 2 months after intraductal injection with MCF7, arrows point to undilated ductal tips (F), MM134, dotted line highlight the subtending duct (G), or SUM44 cells (H), ($n > 6$) scale bar, 1 mm.

I      *Ex vivo* luminescence images of bones, brain, lungs, and liver from MM134 engrafted mice 1 month post-injection (top) and ovaries (bottom) 12 months post-injection. Scale bar, 1 cm.

J      Dot plot showing *ex vivo* bioluminescence of different organs from mice xenografted with MM134 and SUM44 cells plotted over organ and time of analysis. Number of mice analyzed MM134 and SUM44 for 30–180: $N = 5, 10$, 180–270 days: $N = 9, 5$, 270–360 days: $N = 14, 20$, respectively. Dashed line indicates background levels.

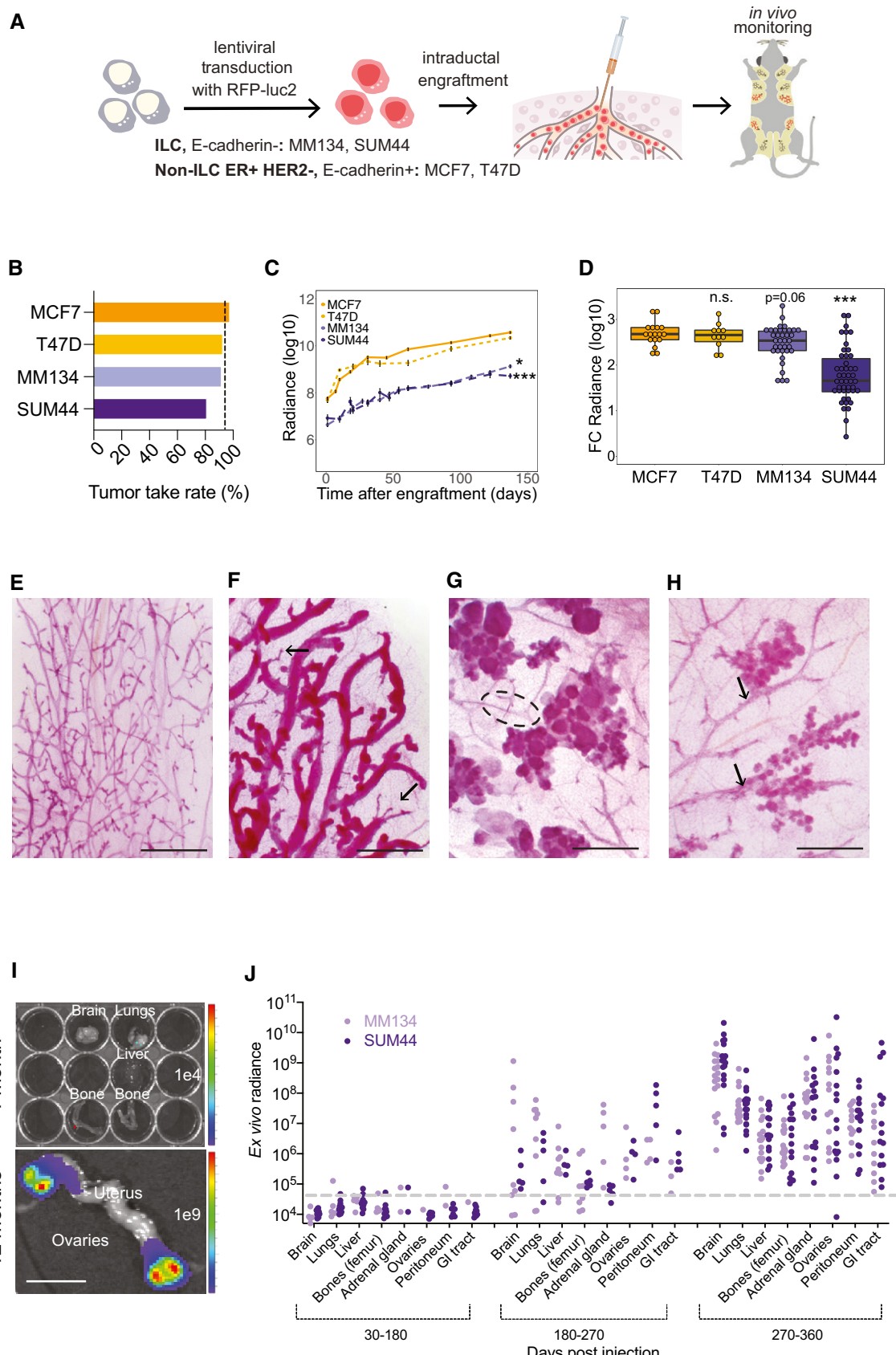

**Figure 1.**

signal in lungs, brains, and bones from mice engrafted with ILC cells (Fig 1I, top, 1J). Both MM134- and SUM44-luc2-RFP cells were first detected at distant sites 5 months after injection. In distinction from MCF7 cells, which metastasized within 2 months of injection, mainly to lungs, bones, liver, and brain (Sflomos et al, 2016), the ILC cells spread more frequently to the adrenal glands, the gastro-intestinal tract, the ovaries (Fig 1I, bottom), and to the peritoneal cavity (Fig 1J). The metastatic load in MCF7 xenografts increased slowly with time due to an increased number of micrometastases. In contrast, in mice xenografted with ILC cells, radiance in distant organs increased to levels comparable to those seen in the primary tumors at 10–12 months after injection and macrometastases appeared (Fig 1I bottom and J). Thus, the intraductal ILC models show the metastatic pattern characteristic of their clinical counter-parts, together with extensive metastatic growth.

## Histopathology of SUM44 and MM134 intraductal xenografts

Histological analysis of the xenografted glands 1 month after injection showed exclusively *in situ* growth of both MM134 (Fig 2A) and SUM44 cells (Fig 2B). At 5 months, invasive growth was observed (Fig 2C and D); in MM134 xenografts, the *in situ* component predominated; non-cohesive, round tumor cells formed groups of nodules, like the "sacs de billes" characteristic of lobular carcinoma *in situ* (LCIS; Fig 2C). The invasive component comprised isolated tumor cells and single file linear cords ("Indian files") (Lakhani, 2012) (Fig 2D, right arrow). SUM44 xenografts were mostly inva-sive; large tumor cells with abundant eosinophilic cytoplasm, pleo-morphic nuclei, numerous mitotic figures, and signet ring cell features were present, reminiscent of pleomorphic ILC (Fig 2D). Microcalcifications previously found in MCF7 intradutctal xeno-grafts were not detected in either of the ILC models.

Immunohistochemistry (IHC) for clinically relevant markers showed that the mean proliferation index was 42.1% in MM134 xenografts with Ki67 heterogeneously expressed, generally higher in the invasive and lower in the *in situ* component (Fig 2E); 61.7% of the tumor cells were ER-positive whereas PR was not detected (Fig 2E). SUM44 xenografts showed more uniform Ki67 labeling with an average index of 30.6% (Fig 2F). Eighty-seven percent of the tumor cells were ER$^+$, whereas PR expression was detected in 11% of the tumor cells (Fig 2F). While the ER status of the xenografts resembled that of their clinical counterparts, PR protein expression was lower. However, stimulation of the host mice with E2 by means of a subcutaneous silastic pellet increased PR index in both models to nearly 35 and 31%, respectively (Fig 2G), indicating that the ER/PR axis is preserved in both models, although MM134 are reported to be PR negative (Christgen & Derksen, 2015).

To better assess the distribution of the ILC cells with respect to the mouse mammary epithelium, we injected the RFP-luc2 tumor cells into NSG-EGFP$^+$ mice (Okabe et al, 1997; Shultz et al, 2005). Double fluorescence stereomicroscopy of engrafted glands 2 months post-intraductal injection revealed multifocal growth characteristic of the human disease (Christgen et al, 2016) (Fig 2H). Immunofluorescence (IF) revealed the GFP$^+$ mouse cells both at the center of the grape-like ILC agglomerates and surrounding them. An antibody that recognizes both human and mouse E-cadherin stained the mouse epithelium in both locations but not the ILC cells (Fig 2I). Co-IF with anti-GFP and α-smooth muscle actin antibodies revealed that the outer cell layer is double positive, indicating that the mouse myoepithelium persists around the *in situ* lesions (Fig 2J). We conclude that ILC cells spread intraepithelially mirroring the pagetoid growth characteris-tic of lobular carcinomas (Balwierz et al, 2014; McCart Reed et al, 2015) (Fig 2K). Thus, the MM134 and SUM44 intraductal xenografts present histopathological features of LCIS and ILC with characteristics of the pleomorphic subtype consistent with their loss of p53 function. Both models recapitulate disease progression *in vivo* with a 4 months *in situ* phase, invasion, and metastasis over 6 months followed by extensive metastatic growth in lungs, meninges, bone, and ovaries (Fig 2K).

## Molecular characteristics of ILC PDXs

Having ascertained that the intraductal microenvironment preserves the distinct biological characteristics of ILC versus non-lobular ER$^+$, HER2$^-$ BC cells, we sought to determine the molecular underpin-nings of these differences using patient-derived carcinoma cells. ILC samples from 15 different patients were obtained, dissociated to single cells, lentivirally transduced with luciferase-GFP and, depend-ing on yields, injected to 4–10 mammary glands of 2–6 recipients. Like the cell line models, patient ILC cells engrafted on average in 80% of the injected glands (Fig 3A).

---

**Figure 2. Histopathology of MM134 and SUM44 intraductal xenografts.**

A, B  Representative micrographs of H&E-stained histological sections of xenografted mammary gland from three female mice 1 month after injection of MM134 (A) or SUM44 cells (B). Scale bars, 100 μm.

C  Representative micrographs of H&E-stained histological sections of MM134 xenografts from three different female mice 5 months after intraductal injection. Arrows point to indian files. Scale bars, 100 μm.

D  Representative micrographs of H&E-stained histological sections of SUM44 cell xenografts from three different female mice 5 months after intraductal injection. Arrows point to indian files. Scale bar, 50 μm.

E, F  Representative micrographs of IHC for Ki67, ER, and PR on histological sections of MM134 (E) and SUM44 (F) xenografts 5 months after intraductal injection counterstained with hematoxylin. Scale bars, 200 μm. Bar plots represent means ± SEM of measurements and indicate the percentage of positive cells for > 1,000 cells counted on three glands.

G  Representative micrographs of PR-IHC on glands from 5-month-old MM134 xenografts from hosts treated for 2 months with vehicle (left) or E2 (right) pellets. Scale bars, 200 μm. Right, bar plot showing percentage of PR$^+$ cells in MM134 and SUM44 xenografts with $n$ > 1,000 cells per condition on glands from three different mice. Data represent mean ± SEM unpaired, Student's $t$-test, two-tailed; ***$P$ < 0.001, ****$P$ < 0.0001.

H  Fluorescence stereo-micrograph of a mammary gland of a *NSG:EGFP$^+$* mouse 2 months after injection with MM134-RFP/luc2 cells. Scale bar, 1 mm.

I, J  Micrographs of IF for GFP and E-cadherin (I) and GFP and SMA (J) counterstained with DAPI of MM134 glands 1 month after intraductal injection of *NSG:EGFP$^+$* mice, $n$ = 4. Scale bars, 50 μm.

K  Scheme showing the timeline of lobular carcinogenesis in the intraductal model.

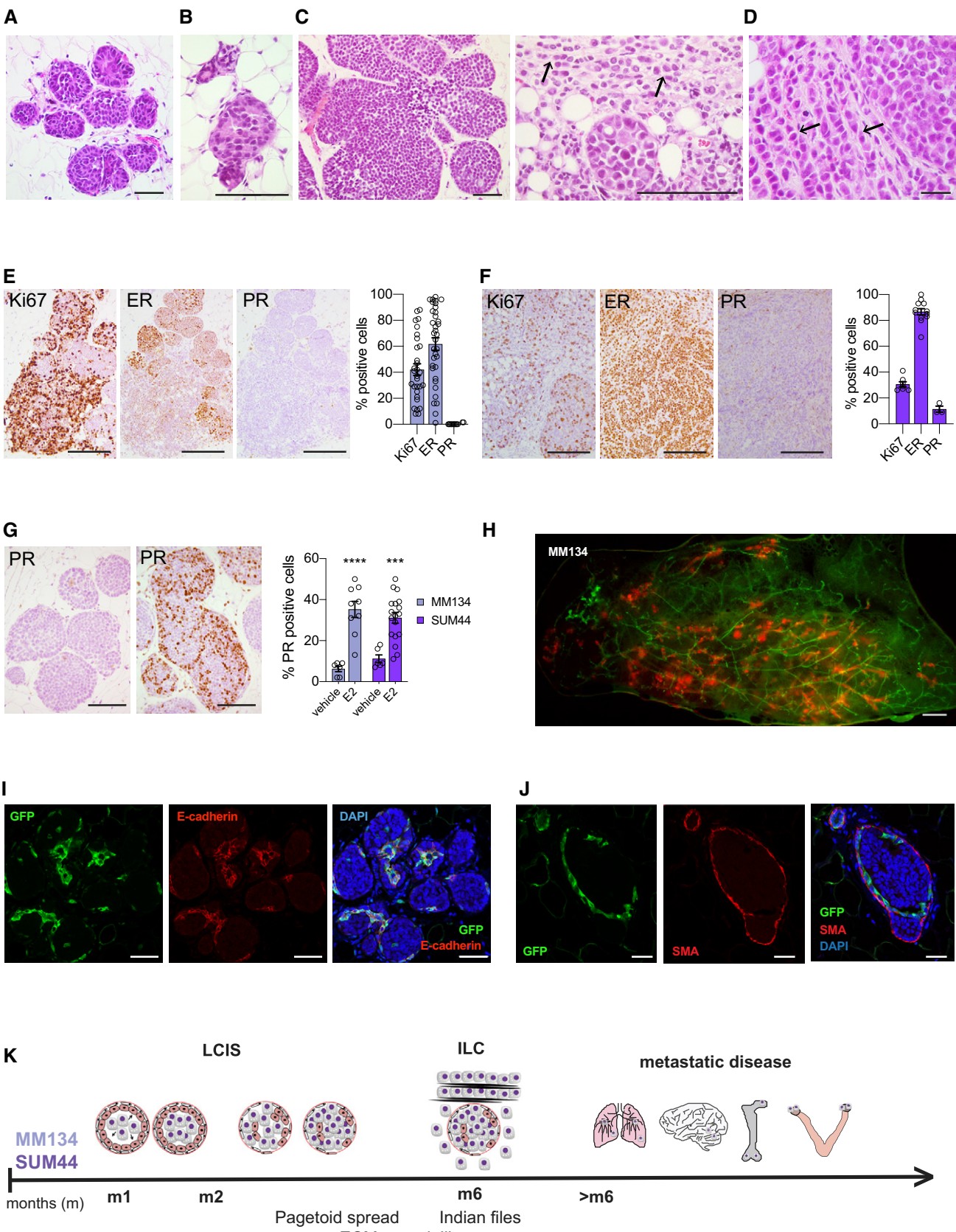

**Figure 2.**

Over 1 year, several PDXs were expanded by serial engraftment and pure tumor cell populations were isolated from three ILC and three non-lobular ER⁺, HER2⁻ BC xenografts by flow cytometry sorting based on GFP expression (Appendix Fig S1A). On average, 20,000 cells were recovered and analyzed by RNA sequencing (Appendix Fig S1B). In contrast to previous global gene expression analyses of tumor biopsies, which compared tumor biopsies (Weigelt *et al*, 2010), pure tumor cells were sampled. Principal component analysis (PCA) showed that the ILC cells were distinct from the non-ILC cells but they did not form a tight cluster (Fig 3B). A total of 1,804 protein-coding genes were differentially expressed (absolute log fold change > 1, *P*-value < 0.05), approximately half of which were increased or decreased (Appendix Fig S1C).

REACTOME pathway analysis of the differentially expressed genes (Fabregat *et al*, 2018) revealed seven biological processes enriched in ILC cells. Consistent with the loss of E-cadherin expression in ILC cells, cell junction, hemidesmosome assembly, and cell-cell communication genes were among the top terms (Fig 3C). Given that only tumor cell-derived RNAs were taken into account, it was surprising to observe that extracellular matrix (ECM) organization ranked as the most enriched gene signature (Fig 3C). Next to collagen type VI alpha 2 chain (*COL6A2*), laminin subunits (*LAMA1* and *LAMA4*) were multiple proteases like matrix metallopeptidase 7 (*MMP7*), cathepsin V (*CTSV*), and several ADAM metallopeptidase with thrombospondin type 1 motif 1 members including *ADAMTS18,* recently shown to be important for mammary stem cell function (Ataca *et al*, 2020) (Fig 3D, E). Interestingly, among the most highly upregulated genes were the luminal progenitor marker *CKIT* (Appendix Fig S1D) and the stem cell marker *ALDH1A3* (Marcato *et al*, 2011).

Gene set enrichment analysis (GSEA) identified gene sets for *lobular* versus *ductal carcinoma* (Appendix Fig S2A) and *E-cadherin target genes* (Appendix Fig S2B), as being significantly enriched in the ILC cells, validating the approach. Consistent with the lower growth rates, cell cycle, G₂/M checkpoint, and E2F targets were down-modulated in the ILC versus the non-ILC ER⁺ HER2⁻ samples (Appendix Fig S2C). Furthermore, MYC targets, MTORC and PI3K/AKT signaling as well as the interferon α response were decreased (Appendix Fig S2D) while the androgen response and TNF-α signaling via NFκB were increased (Appendix Fig S2E).

To identify transcription factor activities that are specific to ILCs, we used the integrated system for motif activity response (ISMARA), which predicts the transcription factors that can potentially govern the differentially expressed genes on the basis of binding motifs at the promoters of these genes (Balwierz *et al*, 2014). The predicted activities of motifs associated with EPAS, BCL3, FOXM1, and TBL1XR1 (Appendix Fig S2F) were higher in ILC than in the non-ILC tumor cells. These transcription factors activate transcription of genes encoding ECM remodeling enzymes and their regulators, structural ECM glycoproteins, as well as proteins structurally or functionally related to collagen deposition including fibrillar collagens ($P < 10^{12}$) (Naba *et al*, 2012) (Fig 3E and F).

In line with the transcriptional profiling, picrosirius red staining of histological sections of both cell line- and patient-derived-xenografts revealed extensive pericellular fibrillar collagen staining (Fig 3G) that was similar to the intratumoral collagen accumulation characteristic of clinical ILC samples (Natal *et al*, 2019).

We have recently shown a link between a protease secreted by epithelial cells, Adamts18, the ECM, and stem cell function in the mouse mammary gland (Ataca *et al*, 2020). In line with this and the concept that ECM determines cell plasticity and cell differentiation (Bruno *et al*, 2017), GSEA revealed a luminal progenitor signature and an increase in basal and mesenchymal versus luminal genes (Fig 3H) suggesting that the special ECM may favor a progenitor-like, de-differentiated cellular state.

## ECM remodeling in patient ILCs

To test whether the ECM remodeling signature identified in the intraductal PDXs is clinically relevant, we interrogated the TCGA breast tumor datasets comprising 127 ILCs and 490 invasive ductal carcinomas (Ciriello *et al*, 2015). Unsupervised hierarchical clustering based on the expression of 44 collagens and 195 glycoprotein-related genes from the matrisome signature (Naba *et al*, 2012) mostly separated the ILCs from the other tumor types (Fig 4A), showing that patient ILCs are enriched for ECM coding genes. Each of the tumors was assigned a score based on the expression of the matrisome signature (Foroutan *et al*, 2018). Compared to non-ILC tumors, ILC samples showed significantly higher scores ($P = 3.4$ e-16) indicating that ECM genes are enriched in ILC than non-ILC tumor cells (Fig 4B). We noticed *COL1A1*, *COL6A1*, *COL14A1*, and *Elastin* (*ELN*) among the most highly enriched transcripts in ILCs (Appendix Fig S3). This begged the question whether ECM remodeling has a role in the genesis of ILCs that might be therapeutically exploited. To address this, we assessed the expression of enzymes, which are responsible for cross-linking of collagens and elastin, the lysyl oxidases (LOX) family (Barker *et al*, 2012) because they can be pharmacologically inhibited. To dispel the possibility the observed

---

**Figure 3. Global gene expression profiles of ILC versus non-ILC ER⁺, HER2⁻ intraductal PDXs.** ▶

A   Bar graph showing take rates in % of PDXs from the treatment naïve patient ILCs. In parentheses number of injected glands (*n*) with growth over total *n* of injected glands. Red dotted line shows average take rate.

B   Principal component analysis plot of global gene expression profiles of ILC and non-ILC ER⁺ HER2⁻ (NST) PDX cells purified by FACS sorting based on GFP expression, *n* = 3.

C   Barplot showing Reactome pathway enrichment analysis of differentially upregulated genes.

D   CNET plot showing network associations between genes and pathways in the top 5 terms with increased expression in ILC.

E   Barplot of top 10 reactome-enriched pathways upregulated in ILC predicted by ISMARA.

F   Barplot of top 10 upregulated MSigDB C2.CP curated canonical pathways collection in ILC PDXs predicted by ISMARA.

G   Representative photomicrographs of picrosirius red stained histological sections of non-ILC (left) and ILC (right) PDXs. Scale bars 100 μm.

H   GSEA plots showing gene sets that are differentially regulated between ILC and non-ILC FACS-sorted intraductal PDXs, related to luminal progenitor features (left) and increased expression of basal and mesenchymal versus luminal genes (middle and right). Significance scores (*P*-values) of GSEAs are computed by nonparametric permutation tests, a procedure that yields only estimates of the *P*-values.

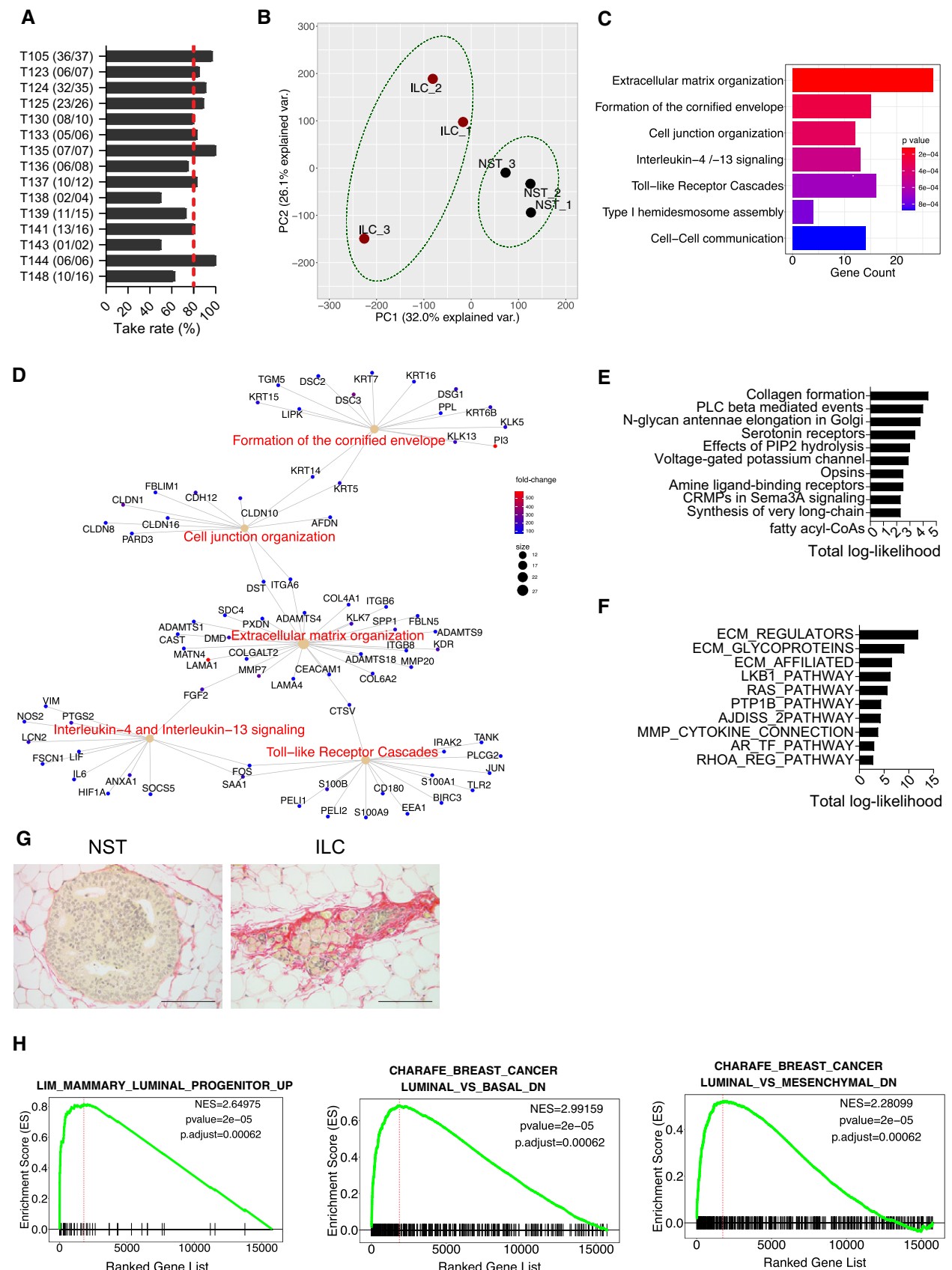

Figure 3.

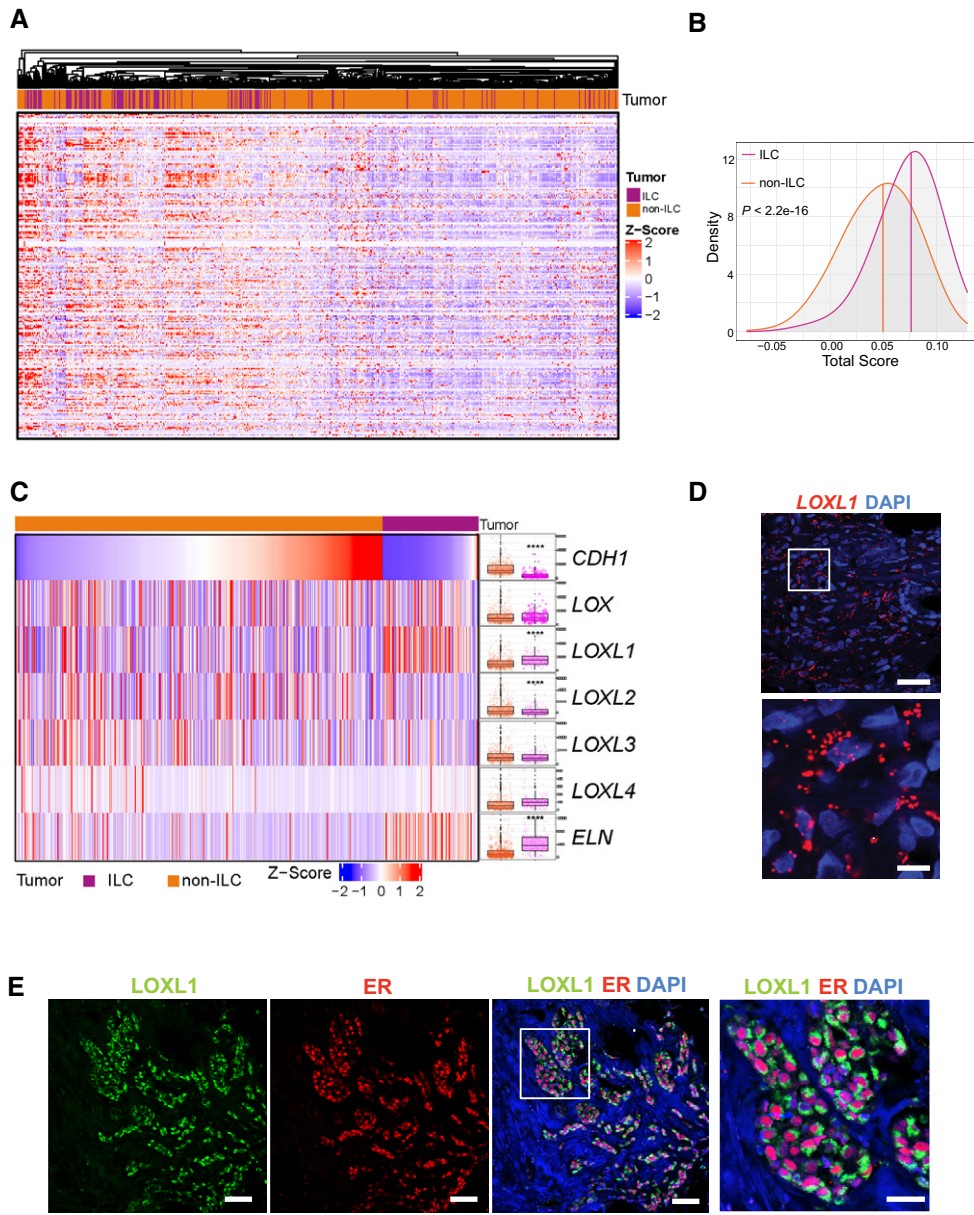

**Figure 4. ECM-related transcriptome in patient samples.**

A  Heatmap of unsupervised hierarchical clustering performed on ER⁺ BC TCGA based on expression of most matrisome collagens and proteoglycans. Tumor identities ILC (blue) and non-ILC (red) are shown.

B  Density plot showing the estimated score distributions of ILC and non-ILC tumors based on the expression of matrisome ECM-related genes. The vertical lines identify the medians. Statistical test performed by unpaired Student's t-test, two-tailed.

C  Heatmap of normalized expression levels of *LOX* family members and *ELN* in invasive BCs, in ILC (blue) and non-ILC (red) tumors sorted based on *CDH1* expression levels. Right, boxplots showing normalized transcript expression levels. The line in the middle of the box is the median. The edges of the box reflect the 25th and 75th percentiles, vertical lines outside the box end at minimum and maximum values. Outliers are depicted as gray dots. Statistical analysis was by ANOVA analysis of the linear model with tumor subtypes and tumor purity as covariates. Adjusted *P*-values computed by post hoc tests. Sample size for NSTs n = 490 and ILCs n = 127. ****P < 0.0001.

D  Representative RNAscope image of sections from patient ILCs (n = 5) stained with anti-*LOXL1* probes (red) and counterstained with DAPI (blue). Scale bars; 50 μm.

E  Representative immunofluorescence image of primary ILC (n = 3) stained with anti-LOXL1 (green) and anti-ER (red) antibodies counterstained with DAPI (blue). Scale bars; 50 μm and right inset 30 μm.

Source data are available online for this figure.

---

differential expression of *LOXL1* was due to differences in tumor purity, ILC versus non-ILC, the cellular composition of the tumor microenvironment was adjusted for (Aran *et al*, 2017). Analysis of

127 ILCs and 490 IDCs (Ciriello *et al*, 2015) showed that, as expected, *CDH1* and *ELN* transcript levels were different in the two groups. The LOX family members, *LOX* and *LOX*-like (*LOXL*) 2–4,

were expressed at comparable levels in ILCs and non-lobular ER$^+$, HER2$^-$ BCs but specifically *LOXL1* transcript levels were different and three-fold higher in ILCs (adjusted *P*-value 4.3 E-10; Fig 4C).

LOXs are reported to be expressed in the tumor stroma, more specifically, in cancer-associated fibroblasts (CAFs) (Trackman, 2016). In light of our finding that the strong ECM signature as well as the increased LOXL1 expression are lobular carcinoma cell-intrinsic, we examined *LOXL1* transcript expression by *in situ* hybridization on ILC samples from seven patients with multiple RNAscope probes. *LOXL1* mRNA was consistently detected in the tumor cells (Fig 4D). Double IF with anti-ER and anti-LOXL1 antibodies showed that the ER$^+$ ILC cells co-stained for LOXL1 (Fig 4E). Thus, LOXL1 is expressed by the lobular carcinoma cells in patient samples.

## Pharmacologic LOX inhibition reduces tumor growth and metastasis

To test whether LOXL1 function is required for tumor growth and progression and may, hence, be exploited as a therapeutic target, we made use of the natural compound β-aminopropionitrile (BAPN). This small molecule is a nucleophilic irreversible inhibitor of the active site of lysyl oxidases (Tang *et al*, 1983). We modeled treatment in two different clinical scenarios. First, we initiated treatment early, 3 weeks after intraductal injection of either SUM44 or MM134 cells, well before xenografts become invasive (Fig 5A). In the case of the MM134 cells, effects of the BAPN treatment appeared only after 6 weeks, whereas in the SUM44 xenografts the treatment showed effects within 3 weeks. Over 3 months, BAPN treatment reduced MM134 and SUM44 xenograft growth by 71 and 75%, respectively (Fig 5A, Appendix Fig S4A). To test the effect of BAPN in the metastatic setting, we initiated treatment 5 months after intraductal injection when disseminated ILC cells can be detected at distant sites. Growth of the MM134 xenografts at the primary site was unaffected by the treatment whereas SUM44 xenograft growth was decreased albeit not as strikingly as by the early treatment (Fig 5B, Appendix Fig S4B). To assess the effect of the drug on metastasis, mice were injected with luciferase prior to sacrifice and

*ex vivo* luminescence of different organs was analyzed. This showed that BAPN treatment reduced metastatic load in lungs, brain, and ovaries of mice engrafted with MM134 cells (Fig 5C). The metastatic load was also decreased in lungs, bones, and ovaries in mice engrafted with SUM44 cells (Fig 5D).

Given the significant enrichment of *LOXL1* expression in ILCs, we hypothesized that the effect of BAPN inhibition is characteristic of ILCs and may not apply to non-ILC ER$^+$, HER2$^-$ BCs. To test this hypothesis, we treated generated intraductal xenografts of the ER$^+$ BC cell line T47D and subjected tumor-bearing mice to BAPN treatment for 60 days. The treatment did not have a significant effect on tumor growth (Fig 5E). Next, we treated two ILC PDXs; while T125 showed a trend, T137 had a significant reduction in tumor growth (Fig 5F). Two non-ILC PDXs showed no response to BAPN treatment (Fig 5G). We note that we chose to use T47D instead of MCF7 because the latter has an amplification of the *LOXL1* locus (Ghandi *et al*, 2019).

Picrosirius red staining of histological sections from advanced tumors revealed abundant fibrillar collagen in the controls and disrupted collagen fibers with dispersed tumor cells in BAPN-treated MM134 xenografts (Fig 5H). Similarly, second-harmonic generation highlighted fibrillar collagen in the control glands and disrupted fibers in the BAPN-treated SUM44 xenografts (Fig 5I).

To identify the molecular changes underlying LOX inhibition, we isolated xenografted SUM44 cells from control and BAPN-treated mice for RNA-seq analysis in the metastatic setting. PCA separated the samples by treatment (Appendix Fig S4C). Over 500 protein-coding genes were differentially expressed (absolute log fold change > 0.5, FDR < 0.05), 223 increased and 368 decreased, upon BAPN treatment (Appendix Fig S4D). REACTOME pathway analysis of the differentially expressed genes revealed seven biological processes negatively affected by the BAPN treatment. Cell cycle-related signaling pathways were among the top terms negatively affected by the treatment (Fig 5J). Consistent with the observed growth inhibition, *MKI67* expression was decreased (Appendix Fig S4E) and GO terms related to cell cycle, DNA repair, mitosis, and progression through G2/M phase were decreased (Appendix Fig S4F–H) similar to what is observed in other tumor types subjected to chemotherapy

---

**Figure 5. Effects of pharmacological LOX inhibition on ILC growth.**

A Experimental design for the early-term treatment: 3 weeks after injection, mice were assigned randomly to vehicle (*n* = 4) or BAPN (*n* = 4) treatment (Top). Radiance-based tumor growth curves (mean ± SE) for the MM134 (left) and SUM44 xenografts (right) treated early (bottom). For statistical analysis, mixed-effects linear models with spline regression (when applicable) were used on log$_{10}$(radiance) curves (Appendix Fig S4A). Likelihood-ratio tests from the ANOVA function in R were used for model comparison, i.e., to check whether the treatment/group covariate is significant. ***P* < 0.001 (likelihood-ratio statistic associated *P*-value).

B Experimental design for treatment in metastatic settings: 5 months after injection, mice were randomized to vehicle (*n* = 4) or BAPN (*n* = 4) treatment (top). Radiance-based tumor growth curves (mean ± SE) for MM134 (left) and SUM44 (right) xenografts treated in the metastatic setting (bottom). For statistical analysis, mixed-effects linear models with spline regression (when applicable) were used on log$_{10}$(radiance) curves (Appendix Fig S4B). Likelihood-ratio tests from the ANOVA function in R were used for model comparison, i.e., to check whether the treatment/group covariate is significant. ***P* < 0.001 (likelihood-ratio statistic associated *P*-value).

C, D Bar graphs showing *ex vivo* radiance ± SEM in different organs harvested from females bearing MM134 or SUM44 intraductal xenografts for the mice described in (B). Statistical significance determined by Student's unpaired *t*-test, two-tailed. *$P$ < 0.05, **$P$ < 0.01, n.s not significant.

E Radiance-based tumor growth curves (mean ± SE) for T47D treated with vehicle (*n* = 8) or BAPN (*n* = 12) 8 days after intraductal injections. n.s not significant.

F Radiance-based tumor growth curves for (mean ± SE) ILC ER$^+$ PDXs T125 and T137 treated with vehicle (*n* = 6, 7) or BAPN (*n* = 7, 8) 66 days after engraftment. **$P$ < 0.01.

G Radiance-based tumor growth curves (mean ± SE) for non-ILC ER$^+$ PDXs T99 and T157 treated with vehicle (*n* = 7, 7) or BAPN (*n* = 8, 12), 8 days after intraductal injection 3 weeks after intraductal injections. n.s not significant.

H Representative micrographs showing picrosirius red staining of sections from MM134 xenografts treated with PBS (left) or BAPN (right), *n* = 4. Scale bars, 100 μm.

I Representative second-harmonic generation (SHG) image on SUM44 xenografts treated with vehicle (left) or BAPN (right), *n* = 3. Collagen fibers in red and DAPI nuclear stain (blue). Scale bar, 50 μm.

J Barplot showing reactome pathway enrichment analysis of differentially down-regulated genes

K GSEA plots showing decreased early and late estrogen response and MYC targets. *P*-values are computed by nonparametric permutation tests.

---

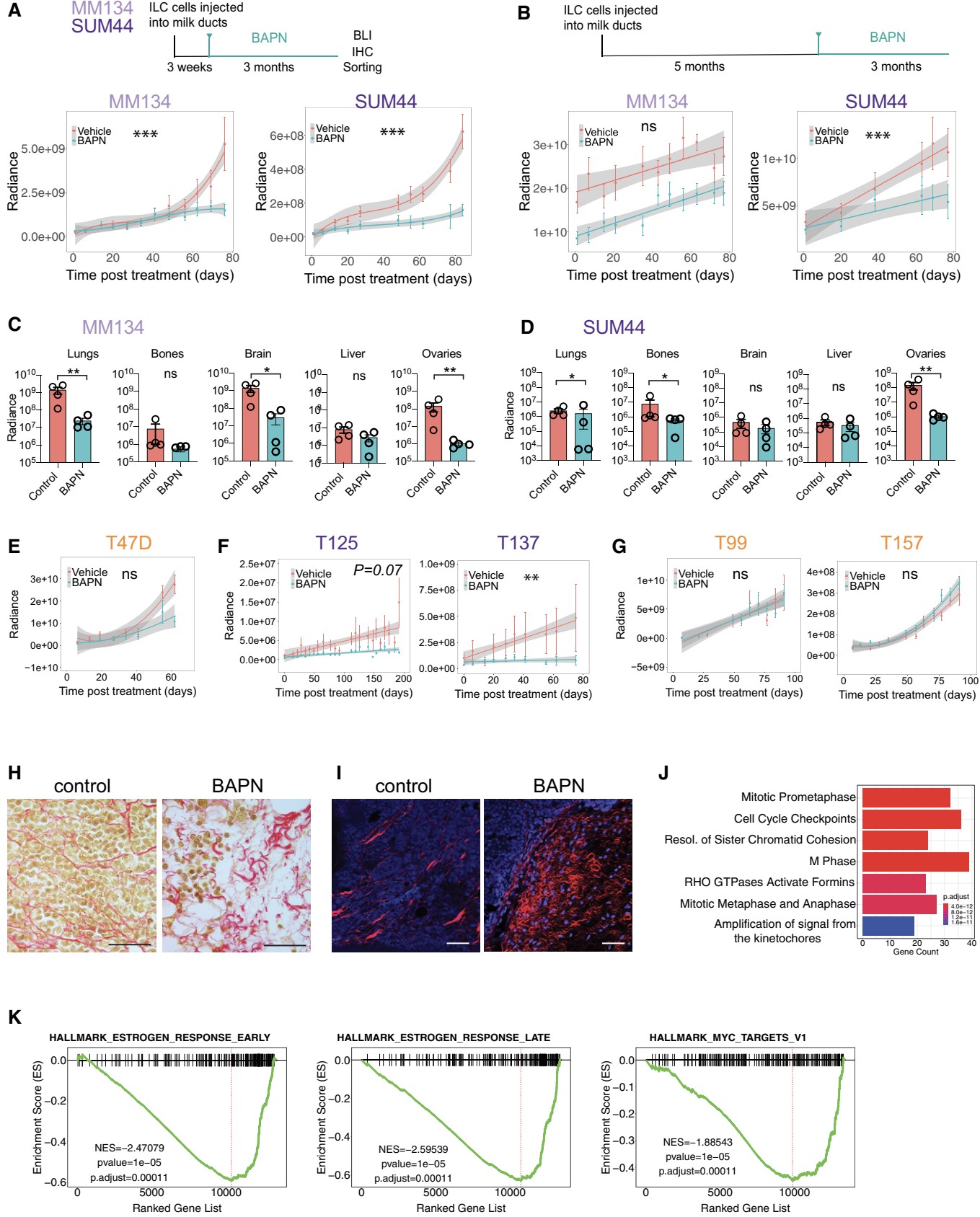

**Figure 5.**

(Creighton *et al*, 2009). GSEA identified gene sets for early and late E2 response and MYC targets as being down-regulated upon BAPN treatment, consistent with estrogen receptor signaling being a major driver of proliferation in these tumors (Fig 5K). Thus, pharmacologic LOX inhibition results in disrupted collagen fibers, decreased estrogen receptor signaling activity, and decreased tumor cell proliferation and tumor progression.

### Role of LOXL1 in ILC progression

BAPN inhibits the activity of all LOX family members, some of which are widely expressed, and has severe side effects precluding clinical use (Alsofi *et al*, 2016). LOXL1, on the other hand, is expressed in only a few tissues, like the eye (Thorleifsson *et al*, 2007), making this family member an attractive drug target. To test whether LOXL1 function is critical or only a minor contributor to the BAPN-induced changes, we adopted a genetic approach and down-modulated specifically *LOXL1* expression in MM134 and SUM44 cells using sh*LOXL1* RNAs. Different sh*LOXL1* constructs were tested and silencing efficiency was around 85% at transcript level as assessed by qRT–PCR (Fig 6A). Next, we infected RFP-luc2 expressing MM134 and SUM44 cells with either *GFP:sh-scramble* or *GFP:shLOXL1* expressing lentivirus. Following FACS sorting for GFP expression (Fig 6B), the cells were engrafted intraductally. Similar to what we observed with the BAPN treatment, in the case of the MM134 cells, the curves began diverging several weeks after engraftment (Fig 6C); by 9 months, luminescence was decreased to 50% of control levels in the sh*LOXL1* infected xenografts (Fig 6C). There was an inhibitory effect of sh*LOXL1* on SUM44 tumor growth in the early phase of intraductal growth, from 4 months onwards the control and sh*LOXL1* curves run in parallel (Fig 6D).

As FACS sorting does not yield 100% pure GFP$^+$ cell populations and *GFP:shLOXL1* expression may inhibit growth, rare uninfected cells or cells in which expression of the viral construct is down-regulated may potentially outgrow. Therefore, in addition to following tumor growth by *in vivo* bioluminescence, we assessed the engrafted mammary glands further at sacrifice. Fluorescence stereomicroscopic analysis of the MM134 xenografted mammary glands (Fig 6E) revealed that the ratio of GFP/RFP signal emanating from glands engrafted with *GFP:sh-scramble* cells was close to 1 and decreased by a third in glands engrafted with *GFP:shLOXL1* cells (Fig 6F) indicating that the presence of sh*LOXL1* bestows a proliferative disadvantage. Despite the advanced stage of the xenografts, the GFP signal in both the control and the sh*LOXL1* infected xenografts emanated mostly from the TDLU-like ductal structure, hence in situ lesions, whereas the RFP-only signal came from larger tumor masses suggesting invasive growth of these cells (Fig 6E). Immunofluorescence staining of sections from these xenografts with an anti-GFP antibody showed that *GFP:sh-scramble* cells were equally distributed over *in situ* and invasive tumor components (Fig 6G left). In contrast, the *GFP:shLOXL1* cells were almost exclusively detected in the *in situ* component (Fig 6G). Thus, in MM134 cells, LOXL1 is required for tumor growth and invasion.

Fluorescence stereomicroscopic analysis of the SUM44 xenografted mammary glands showed mostly intraductal lesions (Fig 6H). While glands engrafted with *GFP:sh-scramble* cells showed comparable amounts of red and green fluorescent signal, the GFP signal was reduced to less than a third of the control levels in the glands engrafted with GFP$^+$ sh*LOXL1* cells (Fig 6H and I). This indicates that the presence of sh*LOXL1* inhibits ILC cell proliferation *in vivo*. Picrosirius red staining of histological sections from sh-*scramble* and sh*LOXL1* ILCs tumors revealed aligned fibrillar collagen in the control tumors and disrupted collagen fibers with dispersed tumor cells in sh*LOXL1* MM134 (Fig 6J) and SUM44 xenografts (Fig 6K) indicative of disrupted collagen fiber remodeling. Thus, while the biological effects are different between the two ILC models, in both cases LOXL1 is essential for *in vivo* tumor progression suggesting that the enzyme is a promising therapeutic target in ILC (Fig 7).

## Discussion

Here we show that intraductal xenografting enables ILC cells to grow *in vivo* with unprecedented take rates. The resulting models

---

**Figure 6. LOXL1 knockdown and its effects on ILC progression.**

A   Bar plot showing *LOXL1* transcript levels as measured by semi qRT–PCR normalized to *36B4* of scramble and sh*LOXL1* expressing ILCs cells *in vitro*. Data represent mean ± SEM (*n* = 3); Student's unpaired *t*-test, two-tailed.

B   Scheme describing the experimental approach used to generate sh*LOXL1* expressing and control ILC xenograft models. ILC:RFP-luc2 cells were transduced with GFP$^+$ *sh-scramble* or GFP$^+$ *shLOXL1* expressing lentiviruses, selected by FACS and injected into the into contralateral 3$^{rd}$ thoracic and the 4$^{th}$ inguinal gland of NSG females. Xenografted mice were monitored for tumor growth by bioluminescence imaging.

C, D   Radiance-based tumor growth curves (mean ± SE) for the scramble and sh*LOXL1* MM134 (C) and SUM44 (D) xenografts. For statistical analysis, mixed-effects with spline regression models were used on log$_{10}$(radiance) curves (Appendix Fig S5A). Likelihood-ratio tests from the ANOVA function in R were used for model comparison, i.e., to check whether the treatment/group covariate is significant. **$P < 0.01$.

E   Representative fluorescent stereo-micrographs of glands injected with MM134:*RFP-luc2* cells (left, top and bottom) infected with either *GFP:sh-scramble* (top) or GFP$^+$ *shLOXL1* (bottom). *n* = 5 and 3. Scale bar, 1,000 μm.

F   Barplot showing the ratio of GFP/RFP signal from fluorescent stereo-micrographs of glands engrafted with MM134 cells, *n* = 8. Statistical analysis by paired Student *t*-test, two-tailed. *$P < 0.05$.

G   Representative fluorescence micrographs of *GFP:sh-scramble* (*n* = 8) and GFP$^+$ *shLOXL1* (*n* = 10) MM134 xenografted glands. ILC cells expressing viral transcripts are detected based on GFP expression (green); DAPI nuclear stain (blue). Asterisks point to intraductal growth while arrow points to invasive tumor cells. Scale bars, 50 μm.

H   Representative fluorescent stereo-micrographs of glands engrafted with SUM44:*RFPluc2* cells infected with either *GFP:sh-scramble* (top) or GFP$^+$ *shLOXL1*, *n* = 6. Scale bar, 1,000 μm.

I   Barplot (mean ± SEM) showing quantification of the ratio GFP/RFP from fluorescent stereo-micrographs of SUM44. Statistical analysis by unpaired Student's *t*-test, two-tailed, *n* = 7 *sh-scramble* and *n* = 6 for *sh-LOXL1*, ****$P < 0.0001$.

J, K   Representative photomicrographs of picrosirius red stained histological sections for *sh-scramble* or *sh-LOXL1* MM134 (J) and SUM44 (K) xenografts. Scale bars, 100 μm.

---

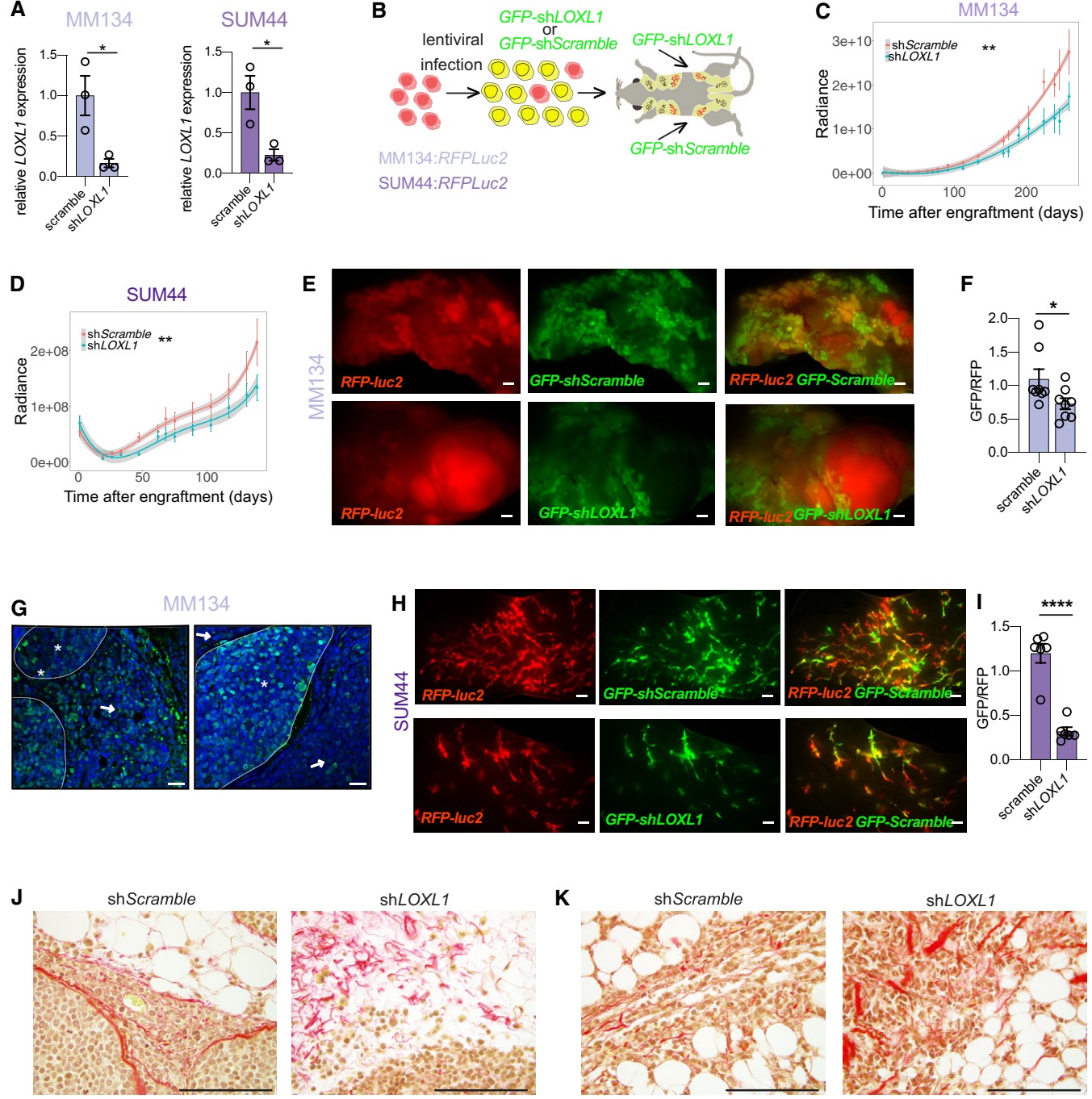

**Figure 6.**

resemble the clinical disease histologically and recapitulate the course of the disease. This opens unique opportunities for basic and translational research into this understudied disease entity and makes critical steps in disease progression, like intraepithelial growth and metastasis finally amenable to experimental exploration. The mechanisms underlying its relative chemoresistance and its unusual response to endocrine therapy can now be studied in a clinically relevant endocrine milieu.

A number of observations provide new insights into the biology of this breast cancer subtype.

Compared to their non-lobular ER$^+$, HER2$^-$ counterparts, both cell line and patient-derived xenografts have lower engraftment rates. We speculate that this is attributable to the lack of E-cadherin expression, because this cell surface receptor is critical for epithelial-epithelial interactions and, hence, important for the adhesion and insertion of these cells into the mouse mammary epithelium.

Upon intraductal engraftment, MCF7 and T47D cells initially grow faster than the ILC cell lines. Strikingly, the former fill and dilate the mouse ducts in a continuous fashion, whereas the ILC cells establish themselves at the ductal tips and form grape-like

structures reminiscent of TDLUs of the human breast. This speaks to the different propagation mode of the lobular cells, which rely on their home-made matrix for proliferation and migration and likely reflects very different interactions with the host's mammary epithelium. The observation that ILC cells home to ductal tips within the mouse milk duct system suggests that these cells are particularly motile and begs the question whether the multifocality characteristic to lobular disease is not a field cancerization phenomenon (Dotto, 2014) but attributable to migration of tumor cells within the ductal tree (Lee *et al*, 2019; Schipper *et al*, 2019).

A limitation of the two models we present here is that both MM134 and SUM44 cells have mutant *TP53*, which is seen only in the pleomorphic lobular carcinomas. The presence of *TP53* mutations in the few available ILC cell lines suggests that the mutation favors *in vitro* establishment. Between PDXs (Fiche *et al*, 2018) and cell line xenografts, the whole spectrum of ILC from LCIS, to classical ILC and pleomorphic ILC is represented. Unlike their clinical counterparts, the cell line xenograft models show low PR expression in line with them being reported as PR⁻ (Christgen & Derksen, 2015). In contrast, PR expression *in vivo* is around the limit of IHC detection and remains E2-inducible. The low basal PR expression may be a consequence of the progesterone level being higher in the host mice than in the postmenopausal women in whom ILCs typically occur, leading to receptor down-modulation (Faivre & Lange, 2007). Alternatively, it may reflect the origin of these cells in pleural effusions (Davidson *et al*, 2004), which frequently have lower PR levels than primary tumors, or be an artifact of adaptation to *in vitro* culture.

The ability to readily purify tumor cells from the new xenograft models enabled us to identify an ILC-specific transcriptional signature that went undetected by global gene expression profiling of tumor biopsies (Du *et al*, 2018), which are confounded by the presence of varying amounts of different stromal cell types. This led to the identification of ECM remodeling as a key ILC feature. The dramatic effects of the treatment with the LOX inhibitor BAPN on ER signaling highlight how this tumor subtype specifically relies on the interaction with the ECM.

We note that the responses both to BAPN and to the effects of sh*LOXL1* are quite different between the two cell lines. This suggests that patient-specific factors need to be taken into account and points to the necessity to develop predictive biomarkers paired with ECM-targeting drugs (Lu *et al*, 2011). LOXL1 overexpression reduced *CDH1* expression and promoted the migration capacity of gastric carcinoma cells and associates LOXL1 with peritoneal dissemination in gastric cancer possibly via promotion of EMT (Hu *et al*, 2020).

BAPN is a small, hydrophilic molecule that is rapidly metabolized into inactive products. It causes lathyrism and high toxicity, likely because it inhibits broad spectrum of lysyl oxidases. In any case, LOXL1-specific inhibitors are an attractive alternative because LOXL1 is distinct from the prototypic lysyl oxidase (LOX) and localizes specifically to sites of elastogenesis where the enzyme interacts with fibulin-5 (Liu *et al*, 2004). Yet, attention to side effects will need to be paid, in particular in the eyes, and the aorta, the two tissues that highly express the enzyme.

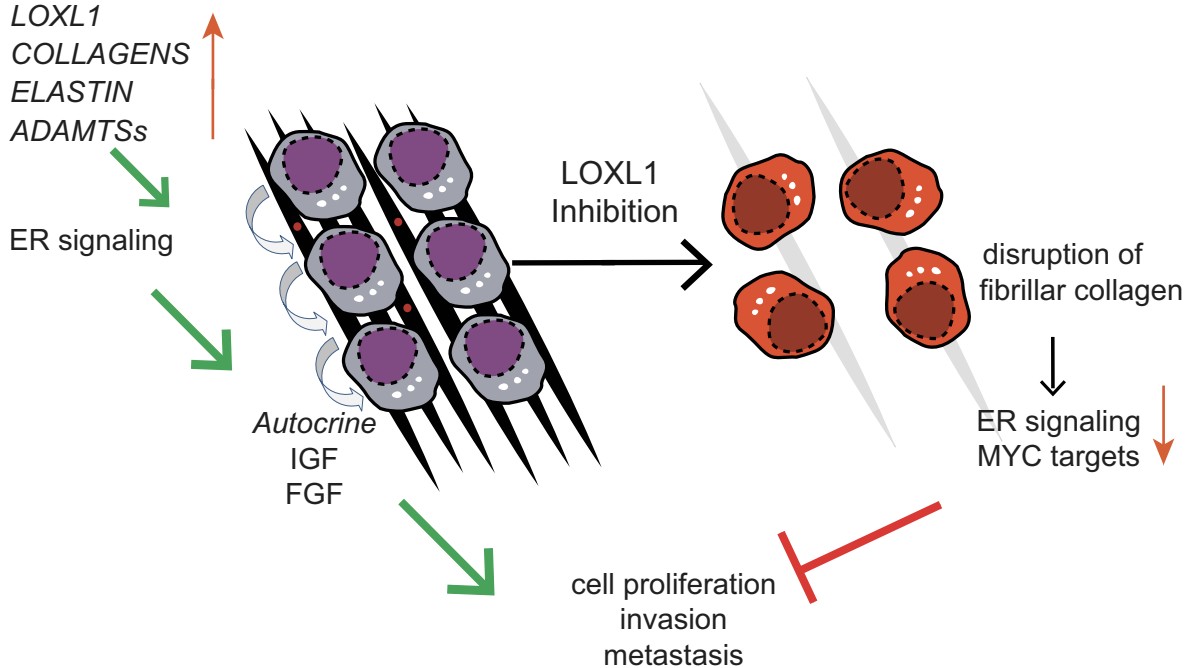

**Figure 7. Model of tumor cell-intrinsic ECM modeling in ILC.**

ILCs are characterized by tumor cell-intrinsic ECM modulating activities, including expression of different collagens, elastin and various secreted proteases like LOXL1 and ADAMTS family members. Cell invasion and metastases is facilitated by the biochemical changes in the ECM and LOXL1 inhibition leads to reduced ER signaling, resulting in decreased survival and diminished proliferation of ILC cells and inhibition of tumor progression.

The effects of *LOXL1* down-modulation point to an important role of the enzyme in tumor invasion and progression, making this enzyme a very attractive drug target. The challenge of defining physiologic substrates will need to be overcome. A better understanding of how LOXL1 is regulated will also be needed in order to anticipate potential mechanisms of resistance.

A synthetic lethal drug screen using the normal basal breast epithelial cell line, MCF10A, with and without E-cadherin identified HDAC, PI3K, and GPCR inhibitors as well as crizotinib an inhibitor of the RTKs c-MET, ALK, and ROS1 (Telford *et al*, 2015; Godwin *et al*, 2019). Extensive genetic and drug screening of many cancer cell lines revealed robust synthetic lethality between E-cadherin loss and specifically inhibition of the orphan receptor tyrosine kinase ROS1 (Bajrami *et al*, 2018). It will be important to assess whether ROS1 and LOXL1 inhibition synergize. Interestingly, E-cadherin loss was associated with increased ROS1 expression (Bajrami *et al*, 2018) and it is tempting to speculate that LOXL1 overexpression in ILC may be causally related to this. The LOXL1 cousin, LOX, was shown to suppress TGFβ1 signaling through a secreted protease HTRA1 in different cell lines models. As a result, matrilin-2 expression was upregulated which resulted in increased EGFR expression on the cell surface (Tang *et al*, 2017). Similar, complex regulation may link LOXL1 to ROS1 and the elucidation of the connection may help to ultimately provide clues to the ligand of this orphan RTK identified more than 30 years ago (Birchmeier *et al*, 1986).

Tissue stiffness resulting from matrix cross-linking is an important factor in tumor progression (Levental *et al*, 2009). Our finding that ECM modulation is a tumor cell-intrinsic feature in ILC is surprising; it is generally held that normal fibroblasts or cancer-associated fibroblasts are the major source of collagen and important modulators of the tumor microenvironment and tissue stiffness. It remains to be investigated whether ECM modulation is a lobular carcinoma-specific trait or it also occurs in other types of tumor cells. Interestingly, global gene expression profiling of ER-deficient mouse mammary epithelial cells revealed that collagen formation is controlled by ER (Cagnet *et al*, 2018). Furthermore, GSEA showed that EMT, stem cell, and Wnt signaling signatures were down in the absence of the ER (Cagnet *et al*, 2018). This points to a tight connection between breast epithelial ER signaling and the ECM. It is tempting to speculate that this aspect of ER signaling activity in turn impinges on tumor cell plasticity and stem cell activity when the ECM has been shown to be critical to the establishment of mammary cell identity in normal mammary gland development (Bruno *et al*, 2017, 1). Our finding that BAPN treatment strikingly reduces ER signaling (Fig 7) suggests that there may be a feedback loop from the ECM to ER signaling that should be explored further and may help elucidate why ILC is relatively tamoxifen resistant (Knauer *et al*, 2015; Metzger Filho *et al*, 2015).

In exploring the possibility that the ECM is the Achilles' Heel of ILC, a better understanding of LOXL1 function, its substrates, and the regulation of its activity are needed. Similarly, the role of other secreted proteases will be interesting to explore. Other members of the ADAMTS family, in particular 5, 8, were previously found upregulated in the ILCs (Ciriello *et al*, 2015), and ADAMTS4, 9, and 18, were found upregulated here. The presented models should assist in discerning these issues.

# Materials and Methods

### Clinical samples

The study was approved by the Commission Cantonale d'éthique de la recherche sur l'être humain (CER-VD 38/15 and PB_2016-01185); informed consent was obtained from all subjects. Experiments conformed to the principles set out in the World Medical Association-Declaration of Helsinki and the Department of Health and the Department of Health and Human Services Belmont Report. After inking of margins and macroscopic assessment by the pathologist, part of the tumor tissue was transported to the laboratory in DMEM/F12 and mechanically and enzymatically digested as previously described. Samples were rinsed and erythrocytes lysed with Red Blood Cell Lysis Buffer (R7757, Sigma) and dissociated to single cells with 0.25% Gibco® Trypsin-EDTA (15400-054, Thermo Fisher Scientific Inc.) for 2 min. Trypsin was inactivated with phosphate buffer saline (PBS) 2% calf serum (CS) followed by incubation with 5 μg/ml deoxyribonuclease DNAase (1284932, Roche AG) in L-15 medium (11415, Gibco) at 37°C for 2 min. 2% CS in PBS was added and the cells were filtered through a 70 μm pore size filter (cat# 352350, BD Falcon) and counted. Patient-derived tumor cells were transduced with bifunctional reporter fusion gene ffLuc2/eGFP lentivirus under control of the cytomegalovirus promoter by spin infection, 25°C for 2.5 h at 400 *g* (Sflomos *et al*, 2016).

### Animal experiments

All the experiments were performed accordance with Swiss guidelines for animal safety. Experiments were performed under protocols 1865.3-5 approved by Service de la Consommation et des Affaires Vétérinaires, Canton de Vaud, Switzerland. NOD.Cg-*Prkdc^scid^ Il2rg^tm1Wjl^*/SzJ and NSG-EGFP (JAX stock #021937) mice were purchased from Jackson Laboratories and further expanded in EPFL with a 12-h-light-12-h-dark cycle, controlled temperature and food and water *ad libitum*. Eight- to 12-week-old female mice were anesthetized by intraperitoneal injection with xylazine 10 mg/kg and ketamine 90 mg/kg (Graeub AG) and intraductally injected into the 3^rd^ and the 4^th^ pair of glands with 4–8 μl of PBS containing 200,000–500,000 cells. Luciferase-based imaging was performed with Xenogen IVIS Imaging System 200 (Caliper Life Sciences) in accordance with the manufacturer's protocols and used to monitor individual mammary glands. At sacrifice, engrafted mammary glands were harvested, fixed in 4% paraformaldehyde for histology and IHC or snap-frozen for RNA and protein isolation. For the pharmacologic inhibition of LOXLs, BAPN (Sigma-Aldrich, A3134) was either injected intraperitoneally at a daily dose of 800 mg/kg or administered with 3 mg/ml BAPN (Sigma-Aldrich, A3134) in the drinking water.

### Bioluminescence imaging

Bioluminescence was detected with Xenogen IVIS Imaging System 200 (Caliper Life Sciences) in accordance with the manufacturer's recommendations and protocols. Ten minutes after intraperitoneal administration of 150 mg/kg luciferin (cat# L-8220, Biosynth AG), images were acquired and analyzed with Living Image software (Caliper Life Sciences, Inc.). For metastasis detection, mice were

injected with 300 μl of luciferin (cat# L-8220, Biosynth AG) 7 min prior to sacrifice, tumors and different organs were dissected within 10–15 min and imaged with IVIS (Perkin Elmer); luminescence was normalized to the average luminescence control organs from ungrafted mice.

### Histology, immunohistochemistry, and immunofluorescence

All samples were fixed in 4% paraformaldehyde and paraffin-embedded (FFPE). For immunostaining, 4-μm sections were mounted onto 76 × 26 mm microscope slides (Rogo-Sampaic, France), deparaffinized in xylene and re-hydrated, antigen retrieval was carried out in 10 mM sodium citrate (pH 6.0) at 95°C for 25 min followed by blocking with 1% BSA for 30 min and overnight incubation with the respective antibodies followed by 1-h incubation with secondary antibodies. Hematoxylin and eosin or sirius red staining were performed according to standard protocols. Sections were counterstained with Mayer's hematoxylin. For fluorescence microscopy, nuclei were counterstained with DAPI (Sigma) then mounted with Fluoromount-GTM (cat# 4958-02, Invitrogen). IF images were acquired on Zeiss LSM700 confocal microscope. The primary and secondary antibodies used are listed in Appendix Table S1. Stereo-micrographs were acquired with M205 FA (Leica). To determine the total area of the GFP/RFP positive gland, filled and not filled, mammary gland was circled using ImageJ software and the percentage of GFP and RFP occupancy was determined as ratio between GFP/RFP areas. GFP and RFP signal intensities were calculated using the following formula: Corrected total cell fluorescence (CTCF) = Integrated density − (Area*Mean fluorescence of background readings).

### Second-harmonic imaging microscopy

Second-harmonic imaging microscopy was performed on a Leica SP5 2-photon microscope with a chameleon ultra laser with pulse width: < 140 fs peak, equipped with two NDD detectors. The collagen channel was excited with a wavelength of 800 nm and detected by a non-descanned detector (NDD1) with a 400/15 nm cutoff filter. The DAPI channel was excited with a wavelength of 800 nm and detected by a non-descanned detector (NDD2) with a 460/50 nm cutoff filter. All images were 12-bit and were acquired with a 20×/ 1.00 HC PL APO water lens (WD 2.00 mm). Image analysis was performed using Leica application suite X and ImageJ.

### Cell culture and gene silencing

MCF7 and T47D (American Type Culture Collection; ATCC) were maintained at 37°C in humidified incubator in an atmosphere of 5% $CO_2$ and grew in Dulbecco's modified Eagle's (DMEM) medium (cat# 31966, Gibco) supplemented with 10% FCS (cat# 10270-106, Thermo Fisher Scientific Inc.) and penicillin/streptomycin (cat# 15070-063, Thermo Fisher Scientific Inc.). MDA-MB-134-VI (ATCC® HTB-23™) and SUM44PE (ASTERAND™) were maintained as described (Ethier *et al*, 1993; Sikora *et al*, 2016). Cells were confirmed to be mycoplasma negative and kept in continuous culture for less than 3 months. For gene silencing, GIPZ microRNA-adapted shRNA lentiviral constructs targeting human *LOXL1* (clones V2LHS_62585 TGAGGATGTAGTTCCCAGG, V3LHS_361814 TTGTT

GCAGAAACGTAGCG, V3LHS_361816 TGCAGAAACGTAGCGACCT, V3LHS_361817 GTAGGTGTCATAGCAGCCT and V3LHS_361818 CGT AGTTCTCGTACTGGCT) (TGCAGAAACGTAGCGACCT, V3LHS_361816) and GIPZ non-silencing shRNA control (RHS4346) were obtained (Dharmacon™ GIPZ™ Lentiviral shRNA). V3LHS_361816 (Fig 6A) showed the highest efficiency and used for the *in vivo* experiments. Lentiviruses were prepared as described (Barde *et al*, 2010).

### RNA *in situ* hybridization

RNAscope assay (Advanced Cell Diagnostics, Cat. No. 323110) was performed according to manufacturer's protocol on 4 μm deparaffinized sections using the following probes: Hs-*LOXL1* (Cat No. 470751), Mm-Ppib (ACD, Cat. No. 313911, positive control) and DapB (ACD, Cat. No. 310043, negative control) at 40°C for 2 h and revealed with TSA Plus-Cy3 (Perkin Elmer, Cat. No. NEL744001KT). Images were captured on confocal Zeiss LSM700.

### RT–PCR

Total RNA was isolated with miRNeasy Mini Kit (Qiagen), cDNA was synthesized with random p(dN)6 primers (Roche) and MMLV reverse transcriptase (Invitrogen). Real-time PCR analysis in triplicates was performed with SYBR Green FastMix (Quanta) reaction mix. Primers used for RT–PCR, see Appendix Table S1.

### RNA-seq experiment

RNA sequencing libraries for the PDXs were prepared by first generating double-stranded cDNA from 10 ng total RNA extracted as described above, from GFP⁺ tumor cells sorted by FACS, with the NuGEN Ovation RNA-Seq System V2 (NuGEN Technologies, San Carlos, California, USA). 100 ng of the resulting double-stranded cDNA were fragmented to 350 bp using Covaris S2 (Covaris, Woburn, Massachusetts, USA). Sequencing libraries were prepared from the fragmented cDNA with the Illumina TruSeq Nano DNA Library Prep Kit (Illumina, San Diego, California, USA) according to the protocol supplied by the manufacturer. Cluster generation was performed with the libraries using the Illumina TruSeq SR Cluster Kit v4 reagents and sequenced on the Illumina HiSeq 2500 with TruSeq SBS Kit v4 reagents. Sequencing data were processed using the Illumina Pipeline Software version 1.82The amplified lanes where added up. After filtering out the low expression tags, TMM normalization was performed in R (Robinson & Oshlack, 2010). A scaling normalization method for differential expression analysis of RNA-seq data was applied (Robinson *et al*, 2010). The differential expression was analyzed using generalized linear models (GLM) in edgeR with a GLM likelihood-ratio test (Robinson *et al*, 2010). For SUM44 RFP⁺ FACS-sorted cells, RNA was extracted using miRNeasy Mini Kit (Qiagen). Libraries were prepared in 2 steps and sequenced on Illumina NextSeq 500 instrument with single-end reads of 85 nt. Base calls and Illumina adaptors trimming performed using bcl2-fastq v2.18. Clontech adaptors trimming performed with CLC 9.

### Computational analysis

Raw reads were aligned to the mouse genome (mm9) using TopHat (v2.0.11)[12]; the exact parameters were tophat -p 6 -g 2

**The paper explained**

**Problem**
Ten to 15% of breast cancers are of a special pathological type called lobular carcinoma. Lobular carcinomas are typically slow growing, very sensitive to hormones and have a high risk of late recurrence and a unique metastatic pattern. They tend not to respond well to standard therapies. Progress in our understanding of this particular type of breast cancer has been hampered by the lack of models to study it.

**Results**
Here, lobular breast cancer cells either from cell lines or from patient tumors are grafted directly to the milk ducts of immunocompromised female mice. We show that in these models, the tumor cells grow, invade, and metastasize in a similar way as they do in patients. Molecular analysis of purified lobular carcinoma cells from intraductal xenografts reveals that these cells actively modulate their extracellular environment. Blocking an enzyme that is critical for this modulation interferes with tumor growth and progression, suggesting that this can be exploited for new therapies.

**Impact**
The new models for lobular carcinoma we have developed and characterized will improve our understanding of the disease. The finding that the lobular tumor cells are highly dependent on the proteins that surround them and that they themselves secrete proteins and enzymes that control this matrix opens new strategies for therapy.

--no-novel-juncs --no-novel-indels --b2- sensitive. Gene counts were generated using FeatureCounts (Liao *et al*, 2014). Differential expression analysis was done using the edgeR package from Bioconductor (Robinson *et al*, 2010). The voom function (Law *et al*, 2014) of the limma package from Bioconductor (Ritchie *et al*, 2015) was used to normalize the data for sequencing depth differences, estimate the mean-variance relationship of the log-counts, and generate a precision weight for each observation so that data were ready for the limma linear fitting function (lmFit). Genes were considered differentially expressed based on *P*-value cutoff ($P < 0.05$). GSEA and REACTOME pathway enrichment analysis were carried out using ClusterProfiler and ReactomePA (Bioconductor), respectively. Default parameters were applied. For GO enrichment analysis, the genome wide annotation for human (org.Hs.eg.db) Bioconductor package as well as the C2 and C5 curated gene set collections from the MSigDB v6.2 (Subramanian *et al*, 2005; Liberzon *et al*, 2011).

The heatmap of matrisome expression was generated using Complex Heatmap package in R. Samples from both non-ILC and ILC samples were sorted according to their *CDH1* normalized counts. Normalized counts are illustrated upon conversion to *Z*-Score according to their values of the rows. The heatmap of *LOXL1* expression in TCGA samples was generated using ComplexHeatmap package in R. Euclidean distance was used to perform unsupervised hierarchical clustering. Normalized counts were converted to *Z*-Score according to the values of the rows. On the side, per each gene included in the heatmap, a boxplot provides the distribution of the expression values of the gene according to the histological subtype of the tumors. Statistical significance was calculated by unpaired Student's *t*-test.

## Statistical analysis

Statistical analyses were performed with GraphPad Prism 8 software and R (nlme and splines packages). Data are shown as means ± SEM, or as otherwise specified. Statistical significance is indicated as follows *$P < 0.05$, **$P < 0.01$, ***$P < 0.001$, ****$P < 0.0001$, • weak significance, n.s. not significant. Tumor purity was derived as described in (Aran *et al*, 2017). Before initiation of treatment, we ensured that control and treatment cohorts have similar radiance values. In case of discrepancy, mice were randomized to reach equal radiance in both cohorts and not blinded to group allocation. Investigators were not blinded to group allocation.

## Data availability

The RNA-seq data produced in this study are available in the Gene Expression Omnibus GSE149671 PDXs (https://www.ncbi.nlm.nih.gov/geo/query/acc.cgi?acc=GSE149671) and GSE149675 SUM44 (https://www.ncbi.nlm.nih.gov/geo/query/acc.cgi?acc=GSE149675).

**Expanded View** for this article is available online.

## Acknowledgements
We thank R. Iggo for critical comments on the manuscript and J. Dessimoz at the EPFL histology core facility, O. Burri and A. Seitz at the EPFL bioimaging and optics platform (BIOP), B. Mangeat at the EPFL gene expression core facility (GECF), M. Garcia V. Glutz and A. Mozes (FCCF) and the Lausanne Genomic Technologies Facility (GTF) for technical assistance. We are grateful to the patients and their families who participated in this study. G.S, V.S., were supported by Biltema ISREC Foundation Cancera Stiftelsen, Mats Paulssons Stiftelse, and Stiftelsen Stefan Paulssons cancerfond; F.DM., by SNF 310030_179163/1 Exploring key steps of the metastatic cascade in ER[+] breast cancer *in vivo*, V.S., P.A. by Swiss Cancer League KFS-3701-08-2015 Lobular carcinoma of the breast:insights from a new PDX model, and L.B by KFS-4738-02-2019-R Different facets of ER signaling during ER[+] breast carcinogenesis.

## Author contributions
GS and CB designed and conceived the study. GS, LB, PA, VS, and AA performed data acquisition. AI-T, RLS, MF, and AS provided administrative, technical, or material support. FDM, PB, and GA analyzed and interpreted the data (e.g., statistical analysis, computational analysis). GS, MF, and CB wrote the manuscript. This work is dedicated to Emanuela (Nela) Busz. Her advocacy voice in Europe helped other patients with lobular breast cancer. Rest in strength.

## Conflict of interest
The authors declare that they have no conflict of interest.

## For more information
Author's websites
- https://www.epfl.ch/labs/brisken-lab/
- https://www.epfl.ch/labs/brisken-lab/preclinicalmodelcourse/
- https://www.cancerprev.com/
- https://cancerprev.ch/

ILC relevant actions
- www.elbcc.org
- https://www.cost.eu/actions/CA19138/#tabs|Name:overview
- https://sumlineknowledgebase.com/?page_id=223

Patients' associations
- https://lobularbreastcancer.org/
- https://lobularireland.com/

Gene list
- *LOXL1* - https://www.ncbi.nlm.nih.gov/gene/4016
- *CDH1* - https://www.ncbi.nlm.nih.gov/gene/999
- *ELN* - https://www.ncbi.nlm.nih.gov/gene/2006
- *ESR1* - https://www.ncbi.nlm.nih.gov/gene/2099
- *PGR* - https://www.ncbi.nlm.nih.gov/gene/5241
- Ki67 - https://www.ncbi.nlm.nih.gov/gene/4288
- *ADAMTS18* - https://www.ncbi.nlm.nih.gov/gene/170692
- *SMA* - https://www.ncbi.nlm.nih.gov/gene/59

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
