## [Review Process File · EMBO Molecular Medicine]

Intraductal xenografts show lobular carcinoma cells rely on their own extracellular matrix and LOXL1

Cathrin Brisken, George Sflomos, Laura Battista, Patrik Aouad, Fabio Demartino, Valentina Scabia, Ayyakkannu Ayyanan, Athina Stravodimou, Assia Ifticene-Treboux, Philipp Bucher, Maryse Fiche, and Giovanna Ambrosini

DOI: [10.15252/emmm.202013180](https://doi.org/10.15252/emmm.202013180)

Corresponding author: Cathrin Brisken (cathrin.brisken@epfl.ch)

Review Timeline:

Submission Date:	26th Jul 20
Editorial Decision:	20th Aug 20
Revision Received:	19th Nov 20
Editorial Decision:	8th Dec 20
Revision Received:	23rd Dec 20
Accepted:	5th Jan 21

Editor: Jingyi Hou

Transaction Report:

20th Aug 2020

Dear Prof. Brisken,

Thank you for the submission of your manuscript to EMBO Molecular Medicine. We have now received feedback from the three referees whom we asked to evaluate your manuscript. As you will see from the reports below, the referees acknowledge the potential interest of the study. However, they also raise a series of concerns about your work, which should be convincingly addressed in a major revision of the present manuscript.

I think that the referees' recommendations are rather clear and there is no need to reiterate their comments. In particular, the referees noted that this manuscript would be improved if the effects of BAPN or LOXi can be tested on MCF7-derived tumors, and if the effects of tamoxifen can be tested on the ILC tumors, which we would strongly encourage you to address to further enhance the impact of the paper. However, if you think that such experiments would take too much time and effort, a discussion of the potential overlap of your findings with DC is required, and the potential limitations in these regards would need to be discussed as well.

We would welcome the submission of a revised version within three months for further consideration. Please note that EMBO Molecular Medicine strongly supports a single round of revision and that, as acceptance or rejection of the manuscript will depend on another round of review, your responses should be as complete as possible.

We are aware that many laboratories cannot function at full efficiency during the current COVID-19/SARS-CoV-2 pandemic and have therefore extended our "scooping protection policy" to cover the period required for a full revision to address the experimental issues. Please let me know should you need additional time, and also if you see a paper with related content published elsewhere.

I look forward to receiving your revised manuscript.

Yours sincerely,

Jingyi Hou

Jingyi Hou
Editor

*** Instructions to submit your revised manuscript ***

** PLEASE NOTE ** As part of the EMBO Publications transparent editorial process initiative (see our Editorial at <https://www.embopress.org/doi/pdf/10.1002/emmm.201000094>), EMBO Molecular Medicine will publish online a Review Process File to accompany accepted manuscripts.

To submit your manuscript, please follow this link:

Link Not Available

- 1) a .docx formatted version of the manuscript text (including Figure legends and tables). Please make sure that the changes are highlighted to be clearly visible to referees and editors alike.
- 2) separate figure files*
- 3) supplemental information as Expanded View and/or Appendix. Please carefully check the authors guidelines for formatting Expanded view and Appendix figures and tables at <https://www.embopress.org/page/journal/17574684/authorguide#expandedview>
- 4) a letter INCLUDING the reviewers' reports and your detailed responses to their comments (as Word file)

Also, and to save some time should your paper be accepted, please read below for additional information regarding some features of our research articles:

- 5) The paper explained: EMBO Molecular Medicine articles are accompanied by a summary of the articles to emphasize the major findings in the paper and their medical implications for the non-specialist reader. Please provide a draft summary of your article highlighting
 - the medical issue you are addressing,
 - the results obtained and
 - their clinical impact.

6) For more information: There is space at the end of each article to list relevant web links for further consultation by our readers. Could you identify some relevant ones and provide such information as well? Some examples are patient associations, relevant databases, OMIM/proteins/genes links, author's websites, etc...

7) Author contributions: the contribution of every author must be detailed in a separate section (before the acknowledgments).

8) EMBO Molecular Medicine now requires a complete author checklist (<https://www.embopress.org/page/journal/17574684/authorguide>) to be submitted with all revised manuscripts. Please use the checklist as a guideline for the sort of information we need WITHIN the manuscript as well as in the checklist. This is particularly important for animal reporting, antibody dilutions (missing) and exact p-values and n that should be indicated instead of a range.

9) Every published paper now includes a 'Synopsis' to further enhance discoverability. Synopses are displayed on the journal webpage and are freely accessible to all readers. They include a short stand first (maximum of 300 characters, including space) as well as 2-5 one sentence bullet points that summarise the paper. Please write the bullet points to summarise the key NEW findings. They should be designed to be complementary to the abstract - i.e. not repeat the same text. We encourage inclusion of key acronyms and quantitative information (maximum of 30 words / bullet point). Please use the passive voice. Please attach these in a separate file or send them by email, we will incorporate them accordingly.

You are also welcome to suggest a striking image or visual abstract to illustrate your article. If you do please provide a jpeg file 550 px-wide x 400-px high.

10) A Conflict of Interest statement should be provided in the main text

11) Please note that we now mandate that all corresponding authors list an ORCID digital identifier. This takes <90 seconds to complete. We encourage all authors to supply an ORCID identifier, which will be linked to their name for unambiguous name identification.

Currently, our records indicate that there is no ORCID associated with your account.

Please click the link below to provide an ORCID:

Link Not Available

12) The system will prompt you to fill in your funding and payment information. This will allow Wiley to send you a quote for the article processing charge (APC) in case of acceptance. This quote takes into account any reduction or fee waivers that you may be eligible for. Authors do not need to pay any fees before their manuscript is accepted and transferred to our publisher.

Each figure should be given in a separate file and should have the following resolution:
Graphs 800-1,200 DPI

Photos 400-800 DPI
Colour (only CMYK) 300-400 DPI"

*Additional important information regarding figures and illustrations can be found at <http://bit.ly/EMBOPressFigurePreparationGuideline>

***** Reviewer's comments *****

Referee #1 (Comments on Novelty/Model System for Author):

There are no outstanding ethical issues or concerns with this manuscript. This is a very good paper - almost publishable as is.

Referee #1 (Remarks for Author):

This manuscript by Sflomos et al. involves characterization of ILC cell line- and tumour-derived xenograft tumours using the innovative intraductal injection system previously published by the same authors. This is an excellent paper, which makes a major contribution to the field. The tumours formed in this study have many features of human ILC, including the correct pattern of organ-specific metastatic dissemination. The tumour cell specific analysis of gene expression in this paper is very informative, and experiment with BAPN, as well as with shLOXL1, are functionally significant. I only have a few very minor suggestions.

1) While its true that mouse models of ILC seem to express lower levels of ER, presentation of work on these models is a little dismissive in tone (for example, the first GEMM model for pILC was published in 2006 making the statement that "understanding the unique ILC biology has been hampered by a lack of models" seem inappropriate). Ideally, caveats with intraductal injection of human ILC cell lines into mice (related to the need for immunodeficient recipient animals, as well as the fact that cell lines being used in this study are models for pleomorphic ILC) can be viewed as orthogonal to caveats with GEMM experiments. This is not meant to be a major criticism and the authors would be better to make no change to the manuscript than to overreact.

2) The schematics for injections in Figure 1 and Figure 6 are a little unclear. Are all mammary glands being injected? Why do glands one and two look yellow/green, whereas glands 3, 4 and 5 are red? Is this meant to be significant? There is one injection arrow in the Figure 1 schematic, whereas in Figure 6 there are two arrows suggesting that LOXL1 knockdown is being performed only on cells injected into glands 1, 2 (maybe 3)(whereas cells injected into glands 4 and 5 are controls)? This should be described better in each figure and also in the methods.

3) The image in Figure 4A is too small to be able to see clustering along the top. Also, it would be nice to use a different colour scheme for tumour type (ILC vs IDC) and expression level in Figure 4C (maybe tumour type could be blue vs yellow or red vs green?)

Referee #2 (Remarks for Author):

Invasive lobular carcinoma (ILC) accounts for 10-15% of breast cancers and is the most common special histological subtype. It has clinical features that are distinct from ductal cancers and the underlying molecular characteristics also differ somewhat from other breast cancers, e.g., consistent loss of E-cadherin which probably contributes to the non-cohesive nature of the tumor cells which grow as 'indian files'. ILCs are often ER/PR positive and a recent review of clinical trials testing various endocrine therapies on ER+ invasive ductal cancers (IDC) and ILCs concludes that response to endocrine agents is similar in the two histological subtypes. In contrast the response to adjuvant chemotherapy is worse for ILC compared to IDC.

While a wealth of models to study DC exist, there are many fewer models of LC which has hindered progress on understanding why ILCs are generally more aggressive despite having a low proliferative index. The manuscript of Sflomos et al describes their work on two ILC -derived cell lines which, when grown as xenografts intraductally in NSG female mice, recapitulate many features of primary human ILCs, including metastatic spread. Using ILC patient derived intraductal xenografts (PDXs) and comparing their gene expression profile to that of non-lobular PDXs, they identified an ECM signature, composed of matrisome genes, that are highly expressed in lobular compared to non-lobular tumors. These are elegant experiments that lead them to propose that there are novel lobular specific targets that could be promising for future therapeutic strategies. The paper should be interesting to many scientists and with further work might be suitable for EMBO Mol Med.

The first figures of the manuscript deal with the characterization of two ER+, E-cadherin negative lobular cancer derived cell lines, MM134 and SUM44, which have been isolated in the past by other groups.

Comment- the fact that they are functionally inactive for p53 should be mentioned in their description.

The cells are xenografted into the mammary ducts, a technology that has been championed by the Briskin group. This technology adds to the impact of the work. There are relevant papers from other labs on these models, but most of the work is with in vitro cell culture (Riggins et al 2008 CR, Stires and Riggins 2018 Mol Cell Endo).

The ILC cells grow more slowly than the MCF7 and T47D ER+ ductal models and they have a slightly lower take rate than the latter. Nevertheless between 80 and 90% of the engrafted glands show tumor growth. Moreover, the tumors metastasize later than the ductal models and compared to data on MCF7 in ref 18, they have a preference for different sites, e.g., adrenal glands and ovaries. Fig 1 also shows that there are major differences in the histology of the LC vs the DC models. The conclusions from Fig 1 & 2 are that both LC cell lines have histological features of LCIS and ILC. Thus, they are a novel in vivo system for studying LC. One caveat is that they both show ER staining, but no PR staining except after stimulating with E2 pellets. This contrast with primary LC which generally expresses both. Potential reasons for this are adequately discussed.

Comment- the model in Fig 2K should be better coordinated with their discussion of the preferred sites of IL vs ductal metastatic disease mentioned on pg 5, and how these differ, comparing LC and ductal models like MCF7.

Work in Fig 3 is devoted to LC patient material. Cells from 15 different patients were tested and found to engraft in about 80% of the intraductal injected glands. They go on to use 3 expanded PDXs from this group to perform RNA seq on tumor cells that were isolated by GFP sorting. The seq data from the ILC PDXs was compared to data from 3 non-lobular ER+ PDXs that were also purified by GFP sorting. What is most novel about the work is that the analyses were carried out on isolated tumor cells, lacking the stroma, so they get a picture of the genes that are tumor specific and might be relevant targets for ILC. The second interesting result is their identification of the biological processes that are enriched in the ILC cells as well as the gene sets differing between the two types. These are mainly summarized in Fig 3 and briefly discussed. The main characteristic that

they concentrate on in the next experiments is the finding that genes encoding ECM remodeling enzymes and regulators are higher in the ILC vs non-ILC tumor cells.

Comment & question- Fig 3G shows an example of collagen staining of ILC and non-ILC human PDXs. Is this something consistently seen in human ILC tumors stained for collagen? Fig 4 A & B, which show data on expression profiling of breast tumors, provides evidence of the matrisome signature that does differ between ILC and other histological sub-types. But this is from intact tumors containing both stroma and tumor cells.

One of their major aims in this paper is to examine if ECM remodeling has a role in ILC growth and might be an exploitable therapeutic target. They concentrated on the lysyl oxidase family of enzymes, which are responsible for cross-linking collagen and elastin, since at least elastin and LOXL1 are significantly increased in expression in ILCs vs non-lobular ER+ breast tumors. Staining for LOXL1 transcripts on tumor cells is shown in Fig 4D. LOXL1 IF is shown in Fig 4E, together with ER staining to detect tumor cells. Thus, tumor cell expression is confirmed. LOXL1 and breast cancer has been studied by the Eler group and shown in breast cancer models, e.g., to play an important role in metastatic spread.

Question- since LOXs are also expressed in fibroblasts, I would have expected to see some staining in ECM cells as well. Can you comment?

The enzymatic activity of LOX enzymes is inhibited by the small molecule compound BAPN. This inhibitor has been tested clinically, but was found to have unacceptable toxicity, probably because it blocks all LOXs, enzymes that are widely expressed.

They tested the potential of BAPN to block growth and metastasis of the 2 ILC models by starting treatment either 3 weeks or 5 months after intraductal injection of the cells (Fig 5). In the first case both models show app 70-75% decrease in primary growth. In the 2nd setting luciferase was measured ex vivo in different organs right after sacrifice. Depending on the model metastatic burden was significantly decreased in different organs. Consistent decreases were seen in lungs and ovaries. (Fig 5C). As expected collagen fibers are disrupted in the tumors of BAPN-treated mice. I find the results from the RNAseq analysis very interesting in particular that genes in the early and late E2 response are down-regulated in the drug treated groups (Fig 5H).

Comments. There is an important experiment that should be added to this analysis. What is the effect of BAPN on the MCF7 ER+ ductal model? Since ductal cancers also express the LOX proteins, it would be important to see if they also respond to the inhibitor. Moreover, by doing an RNA-seq analysis they could see if the effect on E2 response genes is general or specific for the ILC models. Results from this experiment would have an impact on the overall conclusions of the paper and on the model presented in Fig 7.

In the last set of experiments Sflomos et al attempt to link LOXL1 directly to ILC progression by using shLOXL1 RNAs to down-regulate its expression in both ILC models; app 85% downregulation was achieved. FACs sorting of down-regulated cells was achieved via GFP expression sorting. Primary ductal tumor growth was about 50% decreased in the MM134 model, while essentially no impact on primary tumor growth was seen for the SUM44 model, as measured by radiance (Fig 6D). Based on some complicated analysis of RFP and GFP using fluorescent stereomicroscopic analysis they conclude that the 2 models respond differently to KD of LOXL1 but that both models show an effect on invasion, which they use as a measure of progression. It's clear that the KD work is difficult and that the results are not completely conclusive, although the primary growth of MM134 tumors is certainly lower. These data could remain in the paper, but I have a suggestion below.

Comment- Based on the results of Fig5, in particular the finding that genes in the early and late estrogen response category are down-regulated by BAPN, I think that this would be good to follow up on. For example, what happens when the models are treated with an endocrine agent like tamoxifen. At least the SUM44 cells have been described to be tamoxifen sensitive (Sikora et al CR 2014). The clinical data on the benefit of endocrine therapy on patients with ILC is mixed. Thus, having these new models for ILC may be beneficial for further studies on endocrine therapy.

Comment- Finally, the model in Fig 7 needs more work. First, they have LOXL1 inhibition leading to loss of fibrillar fibers. Since the inhibitor affected all LOX enzymes, this should be LOX inhibition. I don't think that fibrillar fibers were checked in LOXL1 KD tumors.

Second, in Fig 7 the ER signature is down-stream of loss of fibrillar fibers, however, they discuss their own work (ref 55) mentioning that in the ER-deficient mammary cells used in that paper, they observe that collagen formation is altered, concluding that its controlled by ER. In conclusion, in ref 55, ER signaling is upstream of collagen expression. Thus, I think it's essential to check the impact of endocrine therapy on in vivo growth and metastasis of ILC models. Potentially, they will need to alter the model in Fig 7.

Comment- The Discussion of the paper presents many interesting diverse thoughts and findings on ILC cancer, however, it is not very streamlined.

Minor changes:

Spelling error on Pg 10 bottom, ampounds, should read amounts.

On pg 13 the statement '...ILC is relatively Tam resistant...' needs a reference.

Referee #3 (Comments on Novelty/Model System for Author):

This manuscript validates a new model for ILC.

Referee #3 (Remarks for Author):

A few years ago this laboratory showed that xenograft transplantation intraductally, rather than into the mammary fat pad, greatly improves the take rate and biological fidelity of ER+ xenografts (reference 18). The current manuscript extends this approach to the Invasive Lobular Carcinoma (ILC) subset of ER+ breast cancers, which have characteristic morphology and suppression of E-Cadherin expression. Two cell lines with these characteristics, MM134 and SUM44 were characterized extensively as intraductal grafts, and found to recapitulate in situ and metastatic features of human ILC including formation of characteristic ILC cords and signet ring cells and lacked microcalcifications seen in MCF-7 grafts. MCF-7 cells tend to fill the ducts, whereas ILC grafts home to the ductal tips. ILC grafts grew and metastasized more slowly than MCF-7, and over extended periods metastasized to adrenal glands, GI tract ovary (as well as lung, brain, and bone) which are less common with MCF-7 and non-ILC. Tumor cells sorted from PDX derived from ILC and non-ILC tumor cells were compared by RNA-seq. Genes upregulated in ILC PDX included genes encoding ECM proteins, consistent with fibrillar collagen accumulation in PDX from ILC. This was accompanied by expression of a less-differentiated, more basal-like luminal progenitor signature. Similarly, ILC in TCGA are enriched for collagen and elastin gene expression. Lysyl oxidase LOXL1, which cross-links collagens and elastin, is upregulated in ILC relative to non-lobular ER+ breast carcinoma. LOXL1 knockdowns reduced growth and progression of SUM44 grafts, and growth of MM134. The non-specific LOX inhibitor BAPN inhibited SUM44 grafts. LOX inhibition disrupted collagen fibers, and also reduced ER signaling as inferred from transcription profiles.

Novelty includes establishing a technique for producing cell line and PDX models that create tumors resembling ER+ ILC, which will have high impact on of this subset of human breast cancer; the surprising importance of tumor cells in ECM remodeling; and the novel therapeutic approach based on transcriptional profiling data implicating ECM changes. The BAPN and knockdown experiments establish proof-of-concept of the novel hypothesis that targeting tissue-selective LOXL1 may contribute to therapeutic response, even though broad targeting with BAPN is not practical because of toxicities.

A few questions below do not detract from the overall quality and impact of this work.

1. The MIND models are based on implantation of tumor cells. What is the fate of donor stromal cells in this model, and is there a way they can be reconstituted along with tumor cells?
2. Are the ILC lines organizing TDLU-like structures, or mechanically filling the ductal tips to which they home?
3. Fig. 1I and 1J. The metastatic patterns of ILC are apparently compared to historical MCF-7 results e.g. from the earlier publication. Have they been directly compared in parallel?
4. Histology was described for MM134 and SUM44 cell lines. Do the PDX recapitulate features of the originating ILC?
5. Do BAPN and LOXL1 knockdown differentially affect ILC vs non-ILC grafts?
6. BAPN has a moderate effect; have the authors combined with Estrogen antagonists?

******* Reviewer's comments *******

Referee #1 (Comments on Novelty/Model System for Author):

There are no outstanding ethical issues or concerns with this manuscript. This is a very good paper - almost publishable as is.

Referee #1 (Remarks for Author):

This manuscript by Sflomos et al. involves characterization of ILC cell line- and tumour-derived xenograft tumours using the innovative intraductal injection system previously published by the same authors. This is an excellent paper, which makes a major contribution to the field. The tumours formed in this study have many features of human ILC, including the correct pattern of organ-specific metastatic dissemination. The tumour cell specific analysis of gene expression in this paper is very informative, and experiment with BAPN, as well as with shLOXL1, are functionally significant. I only have a few very minor suggestions.

We thank this reviewer very much for the appreciation of our work and the positive comments including on the translational aspects of this study as well as the constructive comments.

1) While its true that mouse models of ILC seem to express lower levels of ER, presentation of work on these models is a little dismissive in tone (for example, the first GEMM model for pILC was published in 2006 making the statement that "understanding the unique ILC biology has been hampered by a lack of models" seem inappropriate). Ideally, caveats with intraductal injection of human ILC cell lines into mice (related to the need for immunodeficient recipient animals, as well as the fact that cell lines being used in this study are models for pleomorphic ILC) can be viewed as orthogonal to caveats with GEMM experiments. *This is not meant to be a major criticism and the authors would be better to make no change to the manuscript than to overreact.*

Following the reviewer's comment, we have removed the sentence "Progress in understanding the cellular and molecular mechanisms underlying the unique ILC biology has been hampered by the lack of models. ». We added an additional reference (Derksen et al, 2006), page 3 line 98.

2) The schematics for injections in Figure 1 and Figure 6 are a little unclear. Are all mammary glands being injected? Why do glands one and two look yellow/green, whereas glands 3, 4 and 5 are red? Is this meant to be significant? There is one injection arrow in the Figure 1 schematic, whereas in Figure 6 there are two arrows suggesting that LOXL1 knockdown is being performed only on cells injected into glands 1, 2 (maybe 3)(whereas cells injected into glands 4 and 5 are controls)? This should be described better in each figure and also in the methods.

We have now used the red color only to highlight the milk duct systems of the 3rd (thoracic) and the 4th inguinal glands, which were routinely injected in the present study, **New Figure 1A** and **new Figure 6B**. To better illustrate that the cells infected with scrambled control virus and those infected with shLOXL1 virus were injected into contralateral glands of the same mouse, we have further modified the scheme in the **new Figure 6B**. and arrows illustrate that they have been intraductally injected. We now mention this in the figure legends adapted for Fig.1A line 514 and Fig.6B line 642 as well as in the methods, line 759.

3) The image in Figure 4A is too small to be able to see clustering along the top. Also, it would be nice to use a different colour scheme for tumour type (ILC vs IDC) and expression level in Figure 4C (maybe tumour type could be blue vs yellow or red vs green?)

Following the reviewer's comment, we have corrected the image size in **new Figure 4A** and adapted the color based on the color code for ILC and non-ILC tumors throughout the study also in **new Figure 4B** and **4C**.

Referee #2 (Remarks for Author):

Invasive lobular carcinoma (ILC) accounts for 10-15% of breast cancers and is the most common special histological subtype. It has clinical features that are distinct from ductal cancers and the underlying molecular characteristics also differ somewhat from other breast cancers, e.g., consistent loss of E-cadherin which probably contributes to the non-cohesive nature of the tumor cells which grow as 'indian files'. ILCs are often ER/PR positive and a recent review of clinical trials testing various endocrine therapies on ER+ invasive ductal cancers (IDC) and ILCs concludes that response to endocrine agents is similar in the two histological subtypes. In contrast the response to adjuvant chemotherapy is worse for ILC compared to IDC. While a wealth of models to study DC exist, there are many fewer models of LC which has hindered progress on understanding why ILCs are generally more aggressive despite having a low proliferative index. The manuscript of Sflomos et al describes their work on two ILC -derived cell lines which, when grown as xenografts intraductally in NSG female mice, recapitulate many features of primary human ILCs, including metastatic spread. Using ILC patient derived intraductal xenografts (PDXs) and comparing their gene expression profile to that of non-lobular PDXs, they identified an ECM signature, composed of matrisome genes, that are highly expressed in lobular compared to non-lobular tumors. These are elegant experiments that lead them to propose that there are novel lobular specific targets that could be promising for future therapeutic strategies. The paper should be interesting to many scientists and with further work might be suitable for EMBO Mol Med.

We thank the reviewer for the appreciation of our elegant work and the constructive comments.

The first figures of the manuscript deal with the characterization of two ER+, E-cadherin negative lobular cancer derived cell lines, MM134 and SUM44, which have been isolated in the past by other groups. Comment- the fact that they are functionally inactive for p53 should be mentioned in their description.

We now mention in the introduction that "MM134 and SUM44 "are both functionally inactive for p53 « on page 4, lines 111f and provide two citations.

The cells are xenografted into the mammary ducts, a technology that has been championed by the Brisken group. This technology adds to the impact of the work. There are relevant papers from other labs on these models, but most of the work is with in vitro cell culture (Riggins et el 2008 CR, Stires and Riggins 2018 Mol Cell Endo.)

According to the suggestion of the reviewer, we have cited the findings of Riggins et el 2008 CR, Stires and Riggins 2018 Mol Cell, on page 3 lines 102f.

The ILC cells grow more slowly than the MCF7 and T47D ER+ ductal models and they have a slightly lower take rate than the latter. Nevertheless between 80 and 90% of the engrafted glands show tumor growth. Moreover, the tumors metastasize later than the ductal models and compared to data on MCF7 in ref 18, they have a preference for different sites, e.g., adrenal glands and ovaries. Fig 1 also shows that there are major differences in the histology of the LC vs the DC models. The conclusions from Fig 1 & 2 are that both ILC cells lines have histological features of LCIS and ILC. Thus, they are a novel in vivo system for studying LC. One caveat is that they both show ER staining, but no PR staining except after stimulating with E2 pellets. This contrast with primary ILC which generally expresses both. Potential reasons for this are adequately discussed. Comment- the model in Fig 2K should be better coordinated with their

discussion of the preferred sites of ILC vs ductal metastatic disease mentioned on pg 5, and how these differ, comparing ILC and ductal models like MCF7.

We have better integrated the reference to **new Fig 2K**, page 6 line 204, and now refer in the text to the metastatic organs shown in the scheme, page 6 line 206.

Work in Fig 3 is devoted to LC patient material. Cells from 15 different patients were tested and found to engraft in about 80% of the intraductal injected glands. They go on to use 3 expanded PDXs from this group to perform RNA seq on tumor cells that were isolated by GFP sorting. The seq data from the ILC PDXs was compared to data from 3 non-lobular ER+ PDXs that were also purified by GFP sorting. What is most novel about the work is that the analyses were carried out on isolated tumor cells, lacking the stroma, so they get a picture of the genes that are tumor specific and might be relevant targets for ILC. The second interesting result is their identification of the biological processes that are enriched in the ILC cells as well as the gene sets differing between the two types. These are mainly summarized in Fig 3 and briefly discussed. The main characteristic that they concentrate on in the next experiments is the finding that genes encoding ECM remodeling enzymes and regulators are higher in the ILC vs non-ILC tumor cells. Comment & question- Fig 3G shows an example of collagen staining of ILC and non-ILC human PDXs. Is this something consistently seen in human ILC tumors stained for collagen?

Sirius red staining used in the present study is not routinely used in the clinic. We provide a reference to analysis by Second-Harmonic Generation where the authors characterize the tumor microenvironment of invasive breast cancers of special subtypes including lobular carcinomas. and find that in intra tumoral areas, collagen parameters achieved their lowest values in mucinous, papillary and medullary carcinomas, whereas the highest values were found in classic invasive lobular and tubular carcinomas, line 256.

Fig 4 A & B, which show data on expression profiling of breast tumors, provides evidence of the matrisome signature that does differ between ILC and other histological sub-types. But this is from intact tumors containing both stroma and tumor cells.

One of their major aims in this paper is to examine if ECM remodeling has a role in ILC growth and might be an exploitable therapeutic target. They concentrated on the lysyl oxidase family of enzymes, which are responsible for cross-linking collagen and elastin, since at least elastin and LOXL1 are significantly increased in expression in ILCs vs non-lobular ER+ breast tumors. Staining for LOXL1 transcripts on tumor cells is shown in Fig 4D. LOXL1 IF is shown in Fig 4E, together with ER staining to detect tumor cells. Thus, tumor cell expression is confirmed. LOXL1 and breast cancer has been studied by the Erler group and shown in breast cancer models, e.g., to play an important role in metastatic spread. Question- since LOXs are also expressed in fibroblasts, I would have expected to see some staining in ECM cells as well. Can you comment?

Indeed, based on the literature the reviewer refers to, we were surprised to find *LOXLI* expression in the ILC cells. We note that the in situ data shows *LOXLI* specific probes. *LOXLI* expression is found only in a very limited number of tissues. We think it is likely that other LOX family members are present in the stroma.

The enzymatic activity of LOX enzymes is inhibited by the small molecule compound BAPN. This inhibitor has been tested clinically, but was found to have unacceptable toxicity, probably because it blocks all LOXs, enzymes that are widely expressed. They tested the potential of BAPN to block growth and metastasis of the 2 ILC models by starting treatment either 3 weeks or 5 months after intraductal injection of the cells (Fig 5). In the first case both models show app 70-75% decrease in primary growth. In the 2nd setting luciferase was measured ex vivo in

different organs right after sacrifice. Depending on the model metastatic burden was significantly decreased in different organs. Consistent decreases were seen in lungs and ovaries. (Fig 5C). As expected collagen fibers are disrupted in the tumors of BAPN-treated mice. I find the results from the RNAseq analysis very interesting in particular that genes in the early and late E2 response are down-regulated in the drug treated groups (Fig 5H).

Comments. There is an important experiment that should be added to this analysis. What is the effect of BAPN on the MCF7 ER+ ductal model? Since ductal cancers also express the LOX proteins, it would be important to see if they also respond to the inhibitor. Moreover, by doing an RNA-seq analysis they could see if the effect on E2 response genes is general or specific for the ILC models. Results from this experiment would have an impact on the overall conclusions of the paper and on the model presented in Fig 7.

Following the reviewer's comment, we have now treated non-ILC ER+ HER2- models with BAPN both a cell line, T47D **new Figure 5E**, and 2 PDXs, T99 and T157 **new Figure 5G**. In addition we treated two ILC-PDXs, T125 and T137 **new Figure 5F**. We show that in vivo growth of T47D cells as well as of the 2 non-ILC-PDXs is not affected by the treatment. One ILC-PDX shows a trend and the other ILC-PDX is significantly inhibited. This is now described in the text, page 9, lines 314-321.

We used T47D cells here instead of the MCF7 cells because they are untypical in that they have amplified the LOXL1 gene (Data derived from Broad Institute Cancer Cell Line Encyclopedia (CCLE) and Harmonizome, which provides information about genes and proteins from 114 datasets provided by 66 online resources.

In the last set of experiments, Sflomos et al attempt to link LOXL1 directly to ILC progression by using shLOXL1 RNAs to down-regulate its expression in both ILC models; approx 85% down regulation was achieved. FACS sorting of down-regulated cells was achieved via GFP expression sorting. Primary ductal tumor growth was about 50% decreased in the MM134 model, while essentially no impact on primary tumor growth was seen for the SUM44 model, as measured by radiance (Fig 6D).

Based on some complicated analysis of RFP and GFP using fluorescent stereomicroscopic analysis they conclude that the 2 models respond differently to KD of LOXL1 but that both models show an effect on invasion, which they use as a measure of progression. It's clear that the KD work is difficult and that the results are not completely conclusive, although the primary growth of MM134 tumors is certainly lower. These data could remain in the paper, but I have a suggestion below.

Comment- Based on the results of Fig5, in particular the finding that genes in the early and late estrogen response category are down-regulated by BAPN, I think that this would be good to follow up on. For example, what happens when the models are treated with an endocrine agent like tamoxifen. At least the SUM44 cells have been described to be tamoxifen sensitive (Sikora et al CR 2014). The clinical data on the benefit of endocrine therapy on patients with ILC is mixed. Thus, having these new models for ILC may be beneficial for further studies on endocrine therapy.

We are happy to inform the reviewer that endocrine treatments of the models are ongoing. The preliminary results show an efficient response to estrogen and progesterone depletion as achieved by ovariectomy but MM134 and SUM44 xenografts seem to be slightly stimulated by tamoxifen. However, these experiments last six months and we are not able to add conclusive data at this point.

Comment- Finally, the model in Fig 7 needs more work. First, they have LOXL1 inhibition leading to loss of fibrillar fibers. Since the inhibitor affected all LOX enzymes, this should be LOX inhibition. I don't think that fibrillar fibers were checked in LOXL1 KD tumors.

We thank the reviewer for raising this valid point. We have now checked the collagen fibers on the sh-*LOXI* and control infected ILC tumors, and show that indeed a similar loss of fibrillar collagen arrangement is observed **new figures 6J, K** and in the text, page 11 lines 382-5 and figure legends page 18, lines 616-622.

Second, in Fig 7 the ER signature is down-stream of loss of fibrillar fibers, however, they discuss their own work (ref 55) mentioning that in the ER-deficient mammary cells used in that paper, they observe that collagen formation is altered, concluding that its controlled by ER. In conclusion, in ref 55, ER signaling is upstream of collagen expression. Thus, I think it's essential to check the impact of endocrine therapy on in vivo growth and metastasis of ILC models. Potentially, they will need to alter the model in Fig 7.

We thank the reviewer for pointing to the complex interconnection of ER signaling with collagen. Indeed, in the normal mouse mammary epithelium ER signaling is upstream of collagen expression. However, in the RNAseq analysis of the BAPN treated ILC xenografts we found evidence for down modulation of ER signaling (GSEA, Fig4K). Whether there may be a feedback loop or whether the regulation is altered during tumorigenesis needs will be important to address in the future. Likely the work of other groups such as Linda Schuler's and others interested in stiffness and radiographic density will provide further insights.

Comment- The Discussion of the paper presents many interesting diverse thoughts and findings on ILC cancer, however, it is not very streamlined.

Based on the reviewer's comment we have edited the discussion and added some introductory sentences to clarify the structure.

Minor changes:

Spelling error on Pg 10 bottom, ampounds, should read amounts.

We have corrected the typo, page 11, line 380.

On pg 13 the statement '...ILC is relatively Tam resistant...' needs a reference.

Following the reviewer's comment, we now have added 2 references; Metzger et al., JCO 2015 and Knauer M et al., Abstract S2-06: Survival advantage of anastrozole compared to tamoxifen for lobular breast cancer in the ABCSG-8 study, page 14, lines 485 f.

Referee #3 (Comments on Novelty/Model System for Author):

This manuscript validates a new model for ILC.

Referee #3 (Remarks for Author):

A few years ago this laboratory showed that xenograft transplantation intraductally, rather than into the mammary fat pad, greatly improves the take rate and biological fidelity of ER+ xenografts (reference 18). The current manuscript extends this approach to the Invasive Lobular Carcinoma (ILC) subset of ER+ breast cancers, which have characteristic morphology and suppression of E-Cadherin expression. Two cell lines with these characteristics, MM134 and SUM44 were characterized extensively as intraductal grafts, and found to recapitulate in situ and metastatic features of human ILC including formation of characteristic ILC cords and signet ring cells and lacked microcalcifications seen in MCF-7 grafts. MCF-7 cells tend to fill the ducts, whereas ILC grafts home to the ductal tips. ILC grafts grew and metastasized more slowly than MCF-7, and over extended periods metastasized to adrenal glands, GI tract ovary (as well as lung, brain, and bone) which are less common with MCF-7 and non-ILC. Tumor cells sorted from PDX derived from ILC and non-ILC tumor cells were compared by RNA-seq. Genes upregulated in ILC PDX included genes encoding ECM proteins, consistent with fibrillar

collagen accumulation in PDX from ILC. This was accompanied by expression of a less-differentiated, more basal-like luminal progenitor signature. Similarly, ILC in TCGA are enriched for collagen and elastin gene expression. Lysyl oxidase LOXL1, which cross-links collagens and elastin, is upregulated in ILC relative to non-lobular ER+ breast carcinoma. LOXL1 knockdowns reduced growth and progression of SUM44 grafts, and growth of MM134. The non-specific LOX inhibitor BAPN inhibited SUM44 grafts. LOX inhibition disrupted collagen fibers, and also reduced ER signaling as inferred from transcription profiles.

Novelty includes establishing a technique for producing cell line and PDX models that create tumors resembling ER+ ILC, which will have high impact on of this subset of human breast cancer; the surprising importance of tumor cells in ECM remodeling; and the novel therapeutic approach based on transcriptional profiling data implicating ECM changes. The BAPN and knockdown experiments establish proof-of-concept of the novel hypothesis that targeting tissue-selective LOXL1 may contribute to therapeutic response, even though broad targeting with BAPN is not practical because of toxicities.

We thank the reviewer for positive comments and the interesting questions.

A few questions below do not detract from the overall quality and impact of this work.

1. The MIND models are based on implantation of tumor cells. What is the fate of donor stromal cells in this model, and is there a way they can be reconstituted along with tumor cells?

The ductal microenvironment is highly permissive for luminal cell growth. This seems to be selective as we have failed to date to detect any fibroblasts and/or other stromal cells. We have tested the ability of fibroblasts to grow in the ducts upon intraductal injection without success (Scabia V. et al., unpublished observations).

2. Are the ILC lines organizing TDLU-like structures, or mechanically filling the ductal tips to which they home?

This is a fascinating question. Analysis of glands within a week of injection shows that ILC cells in contrast to non-ILC cells home to ductal tips (Aouad P. et al., unpublished observations). We favor the hypothesis that they are able to organizing TDLU-like structures which means that they can induce morphogenesis.

3. Fig. 1I and 1J. The metastatic patterns of ILC are apparently compared to historical MCF-7 results e.g. from the earlier publication. Have they been directly compared in parallel?

A direct comparison has not been possible as the published study with MCF7 cells was performed in SCID Beige mice. More recently, we injected MCF7 in NSG mice and noticed that MCF7 intraductal tumors are growing faster and the tolerated tumor burden/endpoint is 6-7 months – compared to 9 months for SCID Beige.

4. Histology was described for MM134 and SUM44 cell lines. Do the PDX recapitulate features of the originating ILC?

Most of the ILC-PDXs grow as LCIS or even earlier precursors as intraductal xenografts (Fiche et al. J Pathol 2019, 247(3): 287-292 PMID: 30430577). They retain hormone receptor staining.

5. Do BAPN and LOXL1 knockdown differentially affect ILC vs non-ILC grafts?

Following the reviewer's comment, we have now treated non-ILC ER+ HER2- models with BAPN both a cell line, T47D **new Figure 5E**, and 2 PDXs, T99 and T157 **new Figure 5G**. In addition we treated two ILC-PDXs, T125 and T137 **new Figure 5F**. We show that in vivo growth of T47D cells as well as of the 2 non-ILC-PDXs is not affected by the treatment. One ILC-PDX shows a trend and the other ILC-PDX is significantly inhibited. This is now described in the text, page 9, lines 314-321.

Unfortunately, we were unable to generate and experiment with *shLOXLI* cell lines because the experiments take well over 3 months to perform, see *shLOXLI* experiments in Figure 6C and 6D lasted 250 and 150 days, respectively. We appreciate the reviewer's understanding.

6. BAPN has a moderate effect; have the authors combined with Estrogen antagonists?

We thank the reviewer for this question. We are currently characterizing the response of the ILC models to endocrine therapy and we think that this an excellent suggestion for future studies.

8th Dec 2020

Dear Prof. Brisken,

Thank you for the submission of your revised manuscript to EMBO Molecular Medicine. We have now received the enclosed report from the three referees who were asked to re-assess it. As you will see the referees are all supportive and I am pleased to inform you that we will be able to accept your manuscript pending the following amendments:

1. Please address the minor issue raised by Referee #2 regarding a missing reference.

2. In the main manuscript file, please do the following:

- remove the red color font;
- author contribution: Athina Stravodimou is missing. Also, please use authors' initials.
- Fig 5E, Fig 7 are not called out. There is a callout for Fig S4I but there is no panel I. Please fix.
- RLS from the author list on the title page should be detailed further, e.g. in appendix.
- Please rename "Methods" to "Materials and Methods" and place it after "Discussion".
- in Materials and Methods (and author checklist box #12), include a statement that informed consent was obtained from all subjects and that the experiments conformed to the principles set out in the WMA Declaration of Helsinki and the Department of Health and Human Services Belmont Report.
- in Materials and Methods, for animal work, confirm that all experiments were performed in accordance with relevant guidelines and regulations. The manuscript must include a statement in the Materials and Methods identifying the institutional and/or licensing committee approving the experiments. Gender, age and genetic background must be indicated, along with housing conditions.
- In Materials and Methods, the statistical paragraph should reflect all information that you have filled in the Authors checklist, especially regarding randomization, blinding, replication.

3. source data: provide source data for Fig 4D.

We also encourage you to provide source data for other figures, particularly for electrophoretic gels, blots, but also microscopy images with the aim of making primary data more accessible and transparent to the reader. Would you be willing to provide a PDF file per figure that contains the original, uncropped and unprocessed scans of all or key gels used in the figure (including molecular weight markers)? The PDF files should be labeled with the appropriate figure/panel number (1 file/figure), and should have molecular weight markers; further annotation may be useful but is not essential. The PDF files will be published online with the article as supplementary "Source Data" files. If you have any questions regarding this just contact me.

4. Appendix:

- The Appendix should begin with a short table of contents.
- The legends should be removed from main manuscript file and added to Appendix.
- Nomenclature should be renamed as "Appendix Figure Sx".

5. Rename "Accession Numbers" to "Data availability". Please follow the format below:

Data availability

Also, I cannot find the datasets using the provided accession numbers. Please make sure that the datasets are made publicly available upon acceptance of the manuscript.

6. Our data editor has made a couple of comments on your manuscript (see attached). Please fix these issues and keep the track mode on.

7. The Paper Explained: EMBO Molecular Medicine articles are accompanied by a summary of the articles to emphasize the major findings in the paper and their medical implications for the non-specialist reader. Please provide a draft summary of your article highlighting

8. For more information: There is space at the end of each article to list relevant web links for further consultation by our readers. Could you identify some relevant ones and provide such information as well? Some examples are patient associations, relevant databases, OMIM/proteins/genes links, author's websites, etc...

9. Every published paper now includes a 'Synopsis' to further enhance discoverability. Synopses are displayed on the journal webpage and are freely accessible to all readers. They include a short stand first (maximum of 300 characters, including space) as well as 2-5 one sentence bullet points that summarize the paper. Please write the bullet points to summarize the key NEW findings. They should be designed to be complementary to the abstract - i.e. not repeat the same text. We encourage inclusion of key acronyms and quantitative information (maximum of 30 words / bullet point). Please use the passive voice. Please attach these in a separate file or send them by email, we will incorporate them accordingly.

10. Please also provide a striking image or visual abstract (jpeg format , 550 px-wide x 400-px high) to illustrate your article.

11. As part of the EMBO Publications transparent editorial process initiative (see our Editorial at <http://embomolmed.embopress.org/content/2/9/329>), EMBO Molecular Medicine will publish online a Review Process File (RPF) to accompany accepted manuscripts.

-In the event of acceptance, this file will be published in conjunction with your paper and will include the anonymous referee reports, your point-by-point response and all pertinent correspondence relating to the manuscript. Let us know if you do not agree with this.

I look forward to reading a new revised version of your manuscript as soon as possible.

Sincerely,
Jingyi

Jingyi Hou
Editor
EMBO Molecular Medicine

*** Instructions to submit your revised manuscript ***

To submit your manuscript, please follow this link:

Link Not Available

- 1) a .docx formatted version of the manuscript text (including Figure legends and tables)
- 2) Separate figure files*
- 3) supplemental information as Expanded View and/or Appendix. Please carefully check the authors guidelines for formatting Expanded view and Appendix figures and tables at <https://www.embopress.org/page/journal/17574684/authorguide#expandedview>
- 4) a letter INCLUDING the reviewer's reports and your detailed responses to their comments (as Word file).
- 5) The paper explained: EMBO Molecular Medicine articles are accompanied by a summary of the articles to emphasize the major findings in the paper and their medical implications for the non-specialist reader. Please provide a draft summary of your article highlighting
 - the medical issue you are addressing,
 - the results obtained and
 - their clinical impact.This may be edited to ensure that readers understand the significance and context of the research. Please refer to any of our published articles for an example.

6) For more information: There is space at the end of each article to list relevant web links for further consultation by our readers. Could you identify some relevant ones and provide such information as well? Some examples are patient associations, relevant databases, OMIM/proteins/genes links, author's websites, etc...

7) Author contributions: the contribution of every author must be detailed in a separate section.

8) EMBO Molecular Medicine now requires a complete author checklist (<https://www.embopress.org/page/journal/17574684/authorguide>) to be submitted with all revised manuscripts. Please use the checklist as guideline for the sort of information we need WITHIN the manuscript. The checklist should only be filled with page numbers where the information can be found. This is particularly important for animal reporting, antibody dilutions (missing) and exact values and n that should be indicated instead of a range.

9) Every published paper now includes a 'Synopsis' to further enhance discoverability. Synopses are displayed on the journal webpage and are freely accessible to all readers. They include a short stand first (maximum of 300 characters, including space) as well as 2-5 one sentence bullet points that summarise the paper. Please write the bullet points to summarise the key NEW findings. They should be designed to be complementary to the abstract - i.e. not repeat the same text. We encourage inclusion of key acronyms and quantitative information (maximum of 30 words / bullet point). Please use the passive voice. Please attach these in a separate file or send them by email, we will incorporate them accordingly.

You are also welcome to suggest a striking image or visual abstract to illustrate your article. If you do please provide a jpeg file 550 px-wide x 400-px high.

10) A Conflict of Interest statement should be provided in the main text

11) Please note that we now mandate that all corresponding authors list an ORCID digital identifier. This takes <90 seconds to complete. We encourage all authors to supply an ORCID identifier, which will be linked to their name for unambiguous name identification.

Currently, our records indicate that the ORCID for your account is 0000-0002-6857-3230.

Link Not Available

12) The system will prompt you to fill in your funding and payment information. This will allow Wiley to send you a quote for the article processing charge (APC) in case of acceptance. This quote takes into account any reduction or fee waivers that you may be eligible for. Authors do not need to pay any fees before their manuscript is accepted and transferred to our publisher.

Photos 400-800 DPI

*Additional important information regarding figures and illustrations can be found at <https://bit.ly/EMBOPressFigurePreparationGuideline>

The system will prompt you to fill in your funding and payment information. This will allow Wiley to send you a quote for the article processing charge (APC) in case of acceptance. This quote takes into account any reduction or fee waivers that you may be eligible for. Authors do not need to pay any fees before their manuscript is accepted and transferred to our publisher.

***** Reviewer's comments *****

Referee #1 (Comments on Novelty/Model System for Author):

I think this is an important paper that has been improved upon resubmission. I strongly recommend accepting it in its current form.

Referee #1 (Remarks for Author):

This manuscript has been significantly improved. It represents an important contribution to the field. I recommend acceptance in its current form.

Referee #2 (Comments on Novelty/Model System for Author):

I have nothing to add here. The revised manuscript is excellent on all aspects in my opinion.

Referee #2 (Remarks for Author):

The revised version of the manuscript from the Brisken lab, Sflomos et al, describing the role of the ECM and LOXL1 in malignancy of intraductal PDXs of lobular cancer has gained significantly from the new data as well as some rewriting for clarity. I propose that this version be accepted for publication.

The major changes that I requested were to the most part addressed.

They added some additional discussion and clarification on the p53 status of the cell lines; there is a better integration of the discussion and model in Fig 2K and for Fig 3, more information and a reference on the collagen accumulation in human ILC samples has been added.

An important experiment that I suggested following up on the data shown in Fig 5 was performed. More specifically, the group measured the effect of the BAPN inhibitor on an ER+ ductal cancer model, the T47D breast cancer model and 2 PDXs-T99 and T157. In each case the in vivo growth of these non-ILC models was not affected by the LOX inhibitor (Fig 5 E and G). In addition they added data on 2 new ILC PDXs, T125 & T137, showing that the former had a trend to decreased growth while the latter was significantly blocked by BAPN. This strengthens the fact that blocking LOX enzymes is more important for the ILC tumors.

Minor correction- they should refer to Fig 5E in the text when discussing the results from the T47D

model.

I also suggested another experiment: namely to test the sensitivity of the SUM44 or MM134 cells to the endocrine agent tamoxifen. They mention in the rebuttal letter that this work is ongoing and they have some hints that at least after ovariectomy the ILC models do grow slower. Further analysis of the tam effects is ongoing. I realize that these are long experiments, so I am satisfied with the progress mentioned and hopefully this work will be published in another manuscript once the experiments are finished.

I also asked about the status of the collagen fibers in the LOX1 KD cells and the new Fig 6J-K shows that they are disrupted in both ILC models.

Finally, they made some changes to the Discussion and streamlined their comments. I am satisfied with the changes.

Minor correction- I could not find a reference in the text for Supp Fig S4 A-B. Please add one.

Referee #3 (Comments on Novelty/Model System for Author):

This manuscript validates a new animal model for ILC.

Referee #3 (Remarks for Author):

The authors have made some minor text changes and addressed comments/questions from the reviewers that improve an already strong manuscript.

The authors performed the requested editorial changes.

5th Jan 2021

Dear Prof. Brisken,

We are pleased to inform you that your manuscript is accepted for publication and is now being sent to our publisher to be included in the next available issue of EMBO Molecular Medicine.

We would like to remind you that as part of the EMBO Publications transparent editorial process initiative, EMBO Molecular Medicine will publish a Review Process File online to accompany accepted manuscripts. If you do NOT want the file to be published or would like to exclude figures, please immediately inform the editorial office via e-mail.

Please read below for additional IMPORTANT information regarding your article, its publication and the production process.

Congratulations on your interesting work,

Jingyi Hou

Jingyi Hou
Editor
EMBO Molecular Medicine

Follow us on Twitter @EmboMolMed
Sign up for eTOCs at embopress.org/alertsfeeds

***** Reviewer's comments *****

*** ** IMPORTANT INFORMATION *** **

SPEED OF PUBLICATION

The journal aims for rapid publication of papers, using the advance online publication "Early View" to expedite the process: A properly copy-edited and formatted version will be published as "Early View" after the proofs have been corrected. Please help the Editors and publisher avoid delays by providing e-mail address(es), telephone and fax numbers at which author(s) can be contacted.

Should you be planning a Press Release on your article, please get in contact with embomolmed@wiley.com as early as possible, in order to coordinate publication and release dates.

LICENSE AND PAYMENT:

All articles published in EMBO Molecular Medicine are fully open access: immediately and freely

available to read, download and share.

EMBO Molecular Medicine charges an article processing charge (APC) to cover the publication costs. You, as the corresponding author for this manuscript, should have already received a quote with the article processing fee separately. Please let us know in case this quote has not been received.

Once your article is at Wiley for editorial production you will receive an email from Wiley's Author Services system, which will ask you to log in and will present you with the publication license form for completion. Within the same system the publication fee can be paid by credit card, an invoice, pro forma invoice or purchase order can be requested.

Payment of the publication charge and the signed Open Access Agreement form must be received before the article can be published online.

PROOFS

You will receive the proofs by e-mail approximately 2 weeks after all relevant files have been sent to our Production Office. Please return them within 48 hours and if there should be any problems, please contact the production office at embopressproduction@wiley.com.

Please inform us if there is likely to be any difficulty in reaching you at the above address at that time. Failure to meet our deadlines may result in a delay of publication.

All further communications concerning your paper proofs should quote reference number EMM-2020-13180-V3 and be directed to the production office at embopressproduction@wiley.com.

Thank you,

Jingyi Hou
Editor
EMBO Molecular Medicine

YOU MUST COMPLETE ALL CELLS WITH A PINK BACKGROUND ↓
PLEASE NOTE THAT THIS CHECKLIST WILL BE PUBLISHED ALONGSIDE YOUR PAPER

Corresponding Author Name: Cathrin Brisken, MD, PhD
Journal Submitted to: EMBO Molecular Medicine
Manuscript Number: EMM-2020-13180